Original research

# Development of the Demographic Dividend Effort Index, a novel tool to measure existing efforts to create a favourable environment to harness a demographic dividend: results from an experts' survey from six sub-Saharan African countries

Jean Christophe Rusatira [iD], Carolina Cardona [iD], Michelle Martinez-Baack, Jose G Rimon, Saifuddin Ahmed

Population Family and Reproductive Health, Bloomberg School of Public Health, Johns Hopkins University, Baltimore, Maryland, USA

**Correspondence to**
Dr Jean Christophe Rusatira; jcrusatira@jhu.edu

## ABSTRACT

**Objective** This study aimed to develop a tool to measure the extent of national efforts in policies, services, research and programmes implemented to cultivate and harness the benefits of a potential demographic dividend in six sub-Saharan African countries.

**Design** The survey was self-administered online using the SurveyMonkey platform. The survey questionnaire covered six key sectors: family planning, maternal and child health, education, women's empowerment, labour market, and governance and economic institution. Each sector-specific questionnaire was structured around five practice domains: policymaking, services and programmes, advocacy, research and civil society. Each item was scored from 1 to 10. Factor analysis was used to select the items to be retained for final score estimation. Simple averages were computed to estimate sectoral and domain scores and overall country scores were estimated using weighted country mean scores. Internal consistency, construct validity and reliability were examined using factor analysis and Cronbach's alpha.

**Setting** Sub-Saharan Africa.

**Participants** A total of 440 knowledgeable informants from six countries; namely, Ethiopia (73), Kenya (69), Nigeria (67), Rwanda (54), Senegal (81) and Tanzania (96).

**Results** Based on the results from factor analysis, 38 items were dropped from the analysis and Cronbach's alpha results ranged from 0.84 to 0.98 across domains. The overall demographic dividend effort index (DDEI) scores ranged between 5.4 (95% CI 5.1 to 5.8) in Ethiopia to 7.7 (95% CI 7.5 to 8.0) in Rwanda. In most countries, the disaggregated scores by sector revealed low scores in the labour market and women's empowerment.

**Conclusion** The DDEI scores highlight important gaps in key health and development sectors. The DDEI proved to be a reliable and internally consistent tool for effort measurement in key demographic dividend sectors. The DDEI can serve as a self-evaluation tool for local actors

## STRENGTHS AND LIMITATIONS OF THIS STUDY

⇒ The study questionnaire was developed using existing effort indices and integrated inputs from local partners working in different health and development sectors.
⇒ The study collected data from knowledgeable informants from six countries that are in the early stages of their demographic transition to explore the applicability and utility of a demographic dividend effort index.
⇒ The demographic dividend effort index revealed overall high reliability and internal consistency and validity for measuring the efforts in six key demographic dividend sectors across the six countries.
⇒ The study relied on expert opinions working in different sectors in respective countries, which may be prone to bias and may not be comparable across countries.
⇒ A structural equation model could not be implemented for validation because of sample size limitations.

and may complement existing quantitative tools such as the Global Gender Gap and the Human Capital Index.

## BACKGROUND

The reduction of births and infant deaths during the third stage of the demographic transition produces changes in the age structure of a population.[1] These changes present countries with an opportunity to experience a demographic dividend (DD) characterised by economic growth, sustainable development and social change.[2–4] These demographic changes can spur economic growth by increasing the number of working-age

individuals compared with the number of young dependents. However, favourable policies and investments are required for countries to harness the DD opportunity presented by such population age structure changes.

Across the sub-Saharan African (SSA) region, more than three in four countries are still in the early stages of their demographic transition. They are projected not to reach the peak of the transition, where the DD occurs, until 2050.[5] Although this transition started more than 50 years ago, with declines in fertility levels from an average of 6.8 children per woman to an average of 4.8 children by the mid-2010s, it has been slower compared with other continents.[2] Currently, working-age individuals represent about 60% of the population, and this proportion is expected to keep increasing in the coming few decades.[6] These shifting population dynamics present a unique opportunity for SSA countries to grow their economies and break the cycle of poverty and poor health that has become chronic in the majority of these countries.

Previous literature has identified that, in Asian countries, the DD was responsible for around one-third of their economic growth between 1960s and 1990s, which was possible because a favourable policy environment was put in place.[7 8] These experiences have sparked an interest among African countries. Over the last decade, countries in the region have embarked on the DD development agenda. The majority of the African Union (AU) member states have developed their DD roadmaps and strategic plans.[9] However, there is no standard tool to measure the extent of national efforts in policies, services, research, advocacy and programmes implemented to cultivate and harness the benefits of a potential DD in different SSA countries.

The AU and member states recognise that the DD is a key component to achieving sustainable socioeconomic growth. In 2017, the AU established the theme of the year as 'Harnessing the Demographic Dividend Through Investments in Youth'. This theme generated momentum for the DD across the continent, particularly after the development of the AU DD roadmap. The roadmap identifies labour, education, health, governance and youth empowerment as five fundamental pillars of the DD.[9] Cardona and colleagues further expanded these pillars based on their findings from a systematic literature review that highlighted six key sectors for cultivating and capitalising on the DD potential in SSA.[10] These sectors are governance and economic institutions (GEI), family planning (FP), maternal and child health (MCH), education (ED), women's empowerment (WE) and the labour market (LM).

A central feature to enable a favourable policy environment, and attain a DD, is good governance and strong economic institutions. Governments have the power to attract foreign direct investments to create jobs and contribute to the economy. Strong financial institutions can promote a culture of saving, supported by favourable interest rates to fuel development. Bloom and colleagues have previously documented the correlation between economic stability, institutional quality and economic growth and have evidenced that unstable countries miss out on their DD opportunities.[2 11 12]

Political and economic stability creates an opportunity for a more prosperous community through successful health and development programmes. Successful FP programmes have been proven to improve women's participation in the workforce and have been identified as key interventions to reduce maternal and child mortality due to their impact on birth spacing and fertility levels.[13–16] The reduction of infant mortality is also intertwined with the reduction in fertility and is directly associated with the reduction in maternal mortality. Conley and colleagues evidenced that a reduction of 0.5–0.9 points in the total fertility rate (TFR) was associated with a fall of 50 deaths per 1000 live births in the child mortality rate.[17] This association is concordant with Becker's theory about the quantity–quality trade-off of children, which predicts that parental investments per child increase as fertility levels decline.[18] As families become smaller, more resources are freed to support children's development and the household as a whole.[15 19]

Related to the quantity–quality trade-off, investments in human capital per child increase as fertility lowers, which can accelerate the economy. Ahmed and colleagues found a potential additional 22.4% increase in GDP per capita in SSA from an increase in just 13% in educational attainment between 2011 and 2030, which eventually increases the availability of skilled workers.[20] Equitable access to quality ED for boys and girls is also necessary for countries to generate skilled workers that can later support their economies. Moreover, the ED of girls plays an important role in empowering women, who can later enter the labour force and foster socioeconomic growth in SSA countries.[21]

Africa's young population presents an opportunity for SSA countries to transform their economies. However, to seize this opportunity, countries may require different strategies depending on their demographic transition stage. On the one hand, countries that are still in the initial stage of their demographic transition, also referred to as 'predividend', still need to accelerate their demographic transition and develop the human capital of the growing youth population.[19 22] These countries are characterised by a TFR of four or higher children per woman and a projected growing working-age population by 2030.[19 22] On the other hand, countries that are in the middle of their demographic transition, also referred to as 'early-dividend', require smart investments in human capital development, so that they can fill various job opportunities in the LM. These countries are characterised by having a TFR of fewer than four children per woman and a projected growing working-age population by 2030.[19 22] For the socioeconomic growth to happen, multisectoral efforts are required and countries need to generate jobs faster than their population growth, explore innovative strategies to transition young graduates to formal jobs and facilitate job creation in the private sector.[23]

Over the last four decades, effort measurements have been proven to be reliable approaches to informing international health and development programmes. Since the 1970s, the Family Planning Effort Index (FPEI) has provided reliable estimates to measure the strength of FP programmes in more than 90 countries and has been proven to be valid relative to FP outcomes.[24–26] Building on the success of the FPEI, the same approach has been applied in HIV/AIDS programmes and MCH, which led to the development of the AIDS Program Effort Index (API) in more than 35 countries[27] and the Maternal and Neonatal Program Index (MNPI) in more than 46 countries.[28] However, because health and development systems do not operate in silos, the successful implementation of FPEI, API and MNPI presents an opportunity to develop a multidimensional index to measure DD progress in SSA.

This study aimed to develop an effort assessment index as a standard measure to quantify the nature and strength of in-country DD efforts in the key sectors identified as necessary to harness a DD. This index is meant to be used by national and global actors in the development sector working in five development domains: policy, services and programmes, advocacy, research and civil society.

## METHODOLOGY

### Study design

Data were collected through an online survey that measured the efforts that countries are devoting to maximise their potential to reap the benefits of a DD. The survey was administered through an internet-based modality using the SurveyMonkey platform in six countries: Kenya, Senegal, Nigeria, Rwanda, Tanzania and Ethiopia. These countries were selected because of where they stand in the demographic transition and the convenience to successfully implement the study. The selection of countries sought to include both predividend and early-dividend countries[29] in which the study team had ongoing existing partners with the ability to engage a diverse pool of respondents. The survey(s) was administered in two waves to accommodate COVID-19 lockdowns in different countries. The first wave occurred from August to November 2020, and the second wave that covered Senegal alone took place from March to May 2021. The main language of the questionnaire was English (online supplemental appendix 1), and a French version (online supplemental appendix 2) was created for Senegal by a certified translator.

The questionnaires were developed for each of the six sectors recommended by Cardona and colleagues: FP, MCH, ED, WE, LM and GEI.[10] All questionnaires were structured around five domains: (1) policy; (2) services and programmes; (3) advocacy; (4) research and (5) civil society. These domains were selected based on prior similar effort indices implemented in various health and development areas.[27 28 30 31] These domains also corresponded to the areas of practice in the field and were confirmed by the local partners during the study design

process. The demographic dividend effort index (DDEI) questionnaire was composed of 490 items divided across the six sectors (112 items for FP, 101 items for MCH, 61 items for ED, 71 items for WE, 68 items for LM and 77 items for GEI). Where available, the development of each sector-specific questionnaire integrated validated questions from existing effort indices. For instance, the Family Planning Effort Index (FPEI) questionnaire[30] was integrated into the FP questionnaire. Similarly, specific questions from the MNPI,[28] the Rwanda Governance Scorecard[32] and the AIDS Program Effort Index (API)[27] were integrated into different sector-specific questionnaires. Additional questions were drafted based on DD-favourable interventions for respective sectors as summarised in the wheel of prosperity by Cardona *et al*.[10] More details on the questions drawn from existing indices have been provided as supplemental material (online supplemental appendix 3). During data collection, participants only ever received single surveys that pertained to their sector of expertise and never the entirety of the aggregate survey. Each item was scored on a scale of 1 to 10, from weaker (1) to stronger (10) perceived level of effort or strength. The scoring range of 1–10 has been used in the FPEI and API[27 30] and allowed a wide variability needed to detect differences between domains and sectors.

The DDEI questionnaire was pilot tested among 27 experts from Nigeria, Ethiopia, Tanzania, Senegal, Kenya and Rwanda from June to August 2020. This pilot study allowed local experts to provide their feedback and inputs to the questionnaire, which were integrated into the final revision. The local experts were requested to provide qualitative feedback on content, conciseness and completeness by sector and were requested to provide recommendations on changes to be made. Following the feedback from local experts, the language and content of the questionnaire were amended to improve clarity and conciseness. The final questionnaire was shared with local project partners for final review and approval before administering at the national level across all six countries. Each sector-specific questionnaire included items that assessed the presence and strength of policies, services and programmes, advocacy efforts, research and the involvement of civil society organisations in that sector. No personal identification or sociodemographic characteristics information was collected to limit the concern of being identifiable for the respondents and minimise information bias.

### Study participants

The in-country partners identified the potential participants who were knowledgeable in respective domains and sectors in each country. In-country partners included the Centre d'Excellence Régional en Economie Générationnelle in Senegal, the Center for Communication Programs in Nigeria, the National Council for Population and Development in Kenya, the Tanzania Communication and Development Center, the Ethiopian Academy

of Sciences and the Centre for Integrated Development Research and Action in Rwanda. Participants were considered for the survey if they were regarded to have demonstrated expertise in at least one of the six sectors by the in-country partner. Expertise was defined as having worked in one of the health or development sectors at the national level as a policymaker, advocate, service provider, researcher or member of a civil society organisation. The local study partners had the best knowledge of potential respondents and led the data collection process locally.

As with other similar indices, sampling was necessarily purposive given the need to identify knowledgeable respondents. The design entailed the selection of 10 respondents for each sector. Whenever possible, diversity across the five domains of expertise was prioritised. As such, to reach 10 respondents per sector, each of the six sectors was designed to have two respondents with sector expertise in one of the five domains. These were exclusive classifications such that respondents could only be counted towards the completion of one sector-specific questionnaire. The survey was administered through an online modality, 'SurveyMonkey', in each country. Local partners in respective countries appointed one contact person to respond to any queries regarding the study. Requests to complete the surveys were sent through existing mailing lists of national technical working groups and up to three follow-up calls were made to all individuals for improving survey response rates. The sample size goal was 60 respondents per country.

The expected sample size was on the higher end of sample sizes for similar indices such as the Human Resources Effort Index (N=16–28 per country), the FPEI (N=1–12 per country), the API (N=20–25 per country) and the MNPI (N=10–25).[27 28 30 31] These indices similarly surveyed across various sectors and, sometimes, domains of expertise. However, each of the existing indices was focused on one specific topic, while the DD required a multisectoral sample frame. As this study aimed for 60 respondents per country, considering an expected response rate similar to that of the FPEI of 49%, which was the highest among the other comparative indices, this study ultimately aimed for at least 30 respondents per country.

### Patient and public involvement

No patients or members of the public were involved in the research design, data analysis or dissemination of the findings. Local researchers and policymakers contributed to the design of the questionnaire, the implementation of the survey and were involved in the interpretation of the results.

### Analysis
#### Exploratory data analysis

Exploratory data analysis was done to summarise sample characteristics and distribution of respondents across sectors and domains of expertise at the country level. The proportions of respondents by sector, area of expertise and country were presented in a table to facilitate an easier visualisation.

#### Factor and Cronbach's alpha analysis

Factor analysis was implemented to assess construct internal validity for each domain in each sector. This process allowed us to identify the latent structure revealed by the data. The analysis was conducted by respective sectors retaining one factor for each domain to identify which items loaded on that domain. This process allowed us to assess how each of the items included in the questionnaire loaded on the domain under which the item was administered. Items were retained if their factor loadings were 0.5 or higher. This threshold has been recommended by Costello and Osborne as a more reliable threshold that allows for the selection of items that strongly influence the domain, which improves the internal validity of the measurement.[33] Cronbach's alpha estimates were generated to assess the reliability of the retained items for each domain. A coefficient above 0.60 signals an adequate reliability coefficient.[34]

#### Index scores

The items retained from factor analysis were used to compute mean scores for countries, sectors and domains. Sectoral and domain scores were computed as the simple average across the items retained. The total score for each country was computed as a weighted average that accounts for differences in the number of participants across sectors. Hence, sectoral weights for the total score were constructed as the ratio between the number of participants within a sector and the total number of participants for each country. For all scores, corresponding 95% CIs were calculated with a t distribution because the sample size was relatively small. These estimates were presented graphically for ease of interpretation and comparison across sectors per country.

#### Data translation and availability

To establish policy relevance, the results from this analysis were shared with in-country partners for review and policy translation. The partners from respective countries conducted a series of multisectoral meetings and workshops, in-person and virtual, to generate policy recommendations based on the results from the analysis. The data have been made available through the Harvard Dataverse open-access repository.[35]

#### Heterogeneity testing

Because respondents with different levels of expertise may have systematically assigned different ratings to the same items and the domains of expertise of respondents were differently distributed across countries, we checked for heterogeneity across domains using meta-analytical techniques. This analysis used pooled data by country to allow for sufficient sample size. To assess the overall extent of heterogeneity and whether it was adequate to compute summary estimates, the fixed-effects were compared with random-effect estimates. By using this method, substantial discrepancies between the estimates from the two

**Table 1** Sample characteristics

| | Ethiopia | Kenya | Nigeria | Rwanda | Senegal | Tanzania |
|---|---|---|---|---|---|---|
| | % (Obs.) | % (Obs.) | % (Obs.) | % (Obs.) | % (Obs.) | % (Obs.) |
| Sector | | | | | | |
| Family planning | 23.3 (17) | 24.6 (17) | 26.9 (18) | 13.0 (7) | 8.6 (7) | 39.6 (38) |
| Maternal and child health | 20.5 (15) | 24.6 (17) | 11.9 (8) | 16.7 (9) | 19.8 (16) | 19.8 (19) |
| Education | 20.5 (15) | 7.2 (5) | 17.9 (12) | 13.0 (7) | 13.6 (11) | 10.4 (10) |
| Women's empowerment | 11.0 (8) | 14.5 (10) | 13.4 (9) | 29.6 (16) | 21.0 (17) | 13.5 (13) |
| Labour market | 12.3 (9) | 18.8 (13) | 11.9 (8) | 16.7 (9) | 14.8 (12) | 8.3 (8) |
| Governance and economic institutions | 12.3 (9) | 10.1 (7) | 17.9 (12) | 11.1 (6) | 22.2 (18) | 8.3 (8) |
| Organisation | | | | | | |
| Public sector | 21.9 (16) | 68.1 (47) | 56.7 (38) | 83.3 (45) | 63.6 (49) | 13.5 (13) |
| Private sector | 5.5 (4) | 0.0 (0) | 3.0 (2) | 13.0 (7) | 10.4 (8) | 9.4 (9) |
| Non-governmental organisation | 27.4 (20) | 27.5 (19) | 28.4 (19) | 1.9 (1) | 20.8 (16) | 66.7 (64) |
| University/research institute | 41.1 (30) | 1.4 (1) | 3.0 (2) | 0.0 (0) | 3.9 (3) | 2.1 (2) |
| Other (please specify) | 4.1 (3) | 2.9 (2) | 9.0 (6) | 1.9 (1) | 1.3 (1) | 8.3 (8) |
| Domain | | | | | | |
| Policymaking | 13.7 (10) | 20.3 (14) | 26.9 (18) | 14.8 (8) | 17.3 (14) | 7.3 (7) |
| Services or programmes | 21.9 (16) | 58.0 (40) | 37.3 (25) | 18.5 (10) | 66.7 (54) | 52.1 (50) |
| Advocacy | 8.2 (6) | 11.6 (8) | 13.4 (9) | 14.8 (8) | 8.6 (7) | 25.0 (24) |
| Research | 53.4 (39) | 5.8 (4) | 17.9 (12) | 46.3 (25) | 2.5 (2) | 10.4 (10) |
| Civil society | 2.7 (2) | 4.3 (3) | 4.5 (3) | 5.6 (3) | 4.9 (4) | 5.2 (5) |
| **Obs.** | **73** | **69** | **67** | **54** | **81** | **96** |

Obs, Observations.

models indicate considerable heterogeneity that can make summary estimates misleading.[36] To further assess heterogeneity across domains in respective countries, $I^2$ and $\tau^2$ were used. The $I^2$ values of 25%, 50% and 75% have been interpreted as representing small, moderate and high levels of heterogeneity.[37] $\tau^2$ captures the variance between measurements of efforts by domains which can be captured by random-effect meta-analytic models.

## RESULTS

### Respondents

The sample characteristics of the participants are presented by sector and domain of expertise at the country level (table 1). The total sample consisted of 440 respondents distributed as follows: Ethiopia (73), Kenya (69), Nigeria (67), Rwanda (54), Senegal (81) and Tanzania (96). The number of respondents, domain of expertise and sectors varied across countries. In all countries, except Ethiopia, the majority of respondents were from the public sector and the services and programme domains. The average completion rate was 82% and ranged from 78% in Tanzania to 92% in Rwanda. The intended number of participants per sector was achieved in all six countries.

### Factor and Cronbach's alpha analysis

The items that loaded poorly (<0.5) were dropped from further analysis, which reduced the number of items from 112 to 85 items for FP, 101 to 94 items for MCH, 68 to 65 items for LM and 77 to 76 items for GEI. For detailed factor analysis results by domain and sector, see online supplemental appendix 4. Cronbach's alpha analysis results revealed excellent reliability with alpha estimates ranging from 0.84 in LM to 0.98 in ED (table 2).

### Index scores

The weighted overall country scores were computed to account for differences in the number of participants per sector. For example, in Ethiopia, there were 17 participants from the FP sector, while there were only eight participants in the WE sector.

It would not be a balanced score if it assigned the same weight across sectors. Overall, the weighted average scores ranged from 5.4 (95% CI 5.1 to 5.8) in Ethiopia to 7.7 (95% CI 7.5 to 8.0) in Rwanda (table 3). The second highest scores were for Tanzania (6.3, 95% CI 6.0 to 6.6) and Senegal (6.3, 95% CI 5.9 to 6.7). The second and third lowest scores were for Nigeria (5.5, 95% CI 5.1 to 5.9) and Kenya (5.9, 95% CI 5.5 to 6.2), respectively.

The mean scores per sector were disaggregated by domain of expertise for ease of interpretability and policy

**Table 2** Results from Cronbach's alpha analysis

| | Domain | | | | |
|---|---|---|---|---|---|
| **Sector** | **Policy** | **Services and programmes** | **Advocacy** | **Research** | **Civil society** |
| Family planning | 0.85 | 0.93 | 0.93 | 0.95 | 0.97 |
| Maternal and child health | 0.94 | 0.99 | 0.92 | 0.98 | 0.97 |
| Education | 0.95 | 0.92 | 0.89 | 0.97 | 0.98 |
| Women's empowerment | 0.98 | 0.96 | 0.94 | 0.98 | 0.97 |
| Labour market | 0.95 | 0.95 | 0.84 | 0.96 | 0.97 |
| Governance and economic institutions | 0.95 | 0.97 | 0.94 | 0.97 | 0.94 |

relevance. The total scores disaggregated by sector are presented in figure 1. The radar plots indicate that the sectors with most needs across countries were LM and WE. Both in Nigeria and Ethiopia, the average score for LM and WE struggled to reach the mid-point. However, this was not the case for Rwanda and Tanzania which scored highest in WE, 8.3 (95% CI 8.1 to 8.6) and 6.9 (95% CI 6.1 to 7.8), respectively. Ethiopia was the only country that struggled to reach the mid-point in the GEI sector, 4.9 (95% CI 4.1 to 5.7). By contrast, the MCH, ED and FP sectors received scores above the mid-point across all countries. For example, MCH ranged from a low of 5.1 (95% CI 3.8 to 6.4) in Nigeria to a high of 7.4 (95% CI 7.1 to 7.6) in Rwanda. Furthermore, sectoral scores revealed several interesting variations across domains and countries.

In Ethiopia, across the six sectors, none scored 5 or higher across all five domains. The scores were lowest for civil society engagement relative to other domains in FP, MCH, LM and GEI. In most cases, policy or policy-making scored highest across sectors. Of all the scores, the highest average score was recorded in FP advocacy (6.9, 95% CI 6.0 to 7.9), and the lowest in research for WE (3.8, 95% CI 2.0 to 5.6) as well as civil society engagement

in LM (3.8, 95% CI 2.3 to 5.4). The other scores that were below the mid-point were for services and programmes in WE (4.7, 95% CI 3.3 to 6.1) and LM (4.8, 95% CI 4.1 to 5.4), advocacy in LM (4.7, 95% CI 4.1 to 5.4), research in WE (3.8, 95% CI 2.0 to 5.6) and GEI (4.6, 95% CI 3.6 to 5.6) and civil society engagement in MCH (4.9, 95% CI 3.4 to 6.4), ED (4.0, 95% CI 3.1 to 4.9) and GEI 4.7 (3.5 to 5.8). However, in most instances, the differences in scores between domains were not statistically significant due to overlapping CIs.

In Kenya, the average scores were higher than the mid-point across sectors and domains except for research in LM (4.6, 95% CI 3.7 to 5.4) and civil society in MCH (4.9, 95% CI 3.2 to 6.7). Scores were around 6 for FP in all the domains. Similarly, except for civil society, the scores were around 6 for MCH. The scores for ED, WE and LM were mostly around 5 and hardly reached 6. GEI scored consistently highest compared with the other sectors across the five domains with the highest score in advocacy (7.0, 95% CI 4.5 to 9.5).

In Nigeria, most scores reached the mid-point or higher except for LM which hardly reached 5 in any of the five domains. LM scored lowest in services and programmes (3.7, 95% CI 2.4 to 5.1), followed by policymaking (3.9,

**Table 3** National overall and sector scores

| | Ethiopia | Kenya | Nigeria | Rwanda | Senegal | Tanzania |
|---|---|---|---|---|---|---|
| **Sector** | **Mean (95% CIs) (Obs.)** | | | | | |
| Family planning | 5.8 (5.3 to 6.3) (17) | 6.2 (5.6 to 6.8) (17) | 6.1 (5.2 to 7.0) (18) | 6.5 (6.3 to 6.7) (7) | 6.5 (5.1 to 7.8) (7) | 6.3 (5.9 to 6.7) (38) |
| Maternal and child health | 6.0 (4.7 to 7.2) (15) | 6.2 (5.5 to 7.0) (18) | 5.1 (3.8 to 6.4) (8) | 7.4 (7.1 to 7.6) (9) | 7.2 (6.6 to 7.8) (16) | 6.9 (6.3 to 7.6) (19) |
| Education | 5.3 (4.6 to 6.1) (15) | 5.6 (3.8 to 7.4) (5) | 5.9 (5.1 to 6.8) (12) | 8.2 (7.3 to 9.1) (7) | 5.9 (4.5 to 7.4) (11) | 5.2 (4.1 to 6.4) (10) |
| Women's empowerment | 4.8 (3.4 to 6.1) (8) | 5.1 (4.2 to 5.9) (10) | 5.1 (4.1 to 6.1) (9) | 8.3 (8.1 to 8.6) (16) | 6.4 (5.6 to 7.3) (17) | 6.9 (6.1 to 7.8) (13) |
| Labour market | 5.1 (4.3 to 5.9) (9) | 5.3 (4.7 to 5.9) (13) | 4.2 (2.9 to 5.5) (8) | 7.6 (7.3 to 7.8) (9) | 5.2 (3.7 to 6.7) (12) | 5.7 (4.3 to 7.2) (8) |
| Governance and economic institutions | 4.9 (4.1 to 5.7) (9) | 6.5 (4.6 to 8.5) (6) | 5.7 (4.7 to 6.7) (12) | 7.8 (7.2 to 8.4) (6) | 6.3 (5.7 to 6.9) (18) | 5.4 (4.3 to 6.6) (8) |
| Weighted overall score | 5.4 (5.1 to 5.8) (73) | 5.9 (5.5 to 6.2) (69) | 5.5 (5.1 to 5.9) (67) | 7.7 (7.5 to 8.0) (54) | 6.3 (5.9 to 6.7) (81) | 6.3 (6.0 to 6.6) (96) |

Obs, Observations.

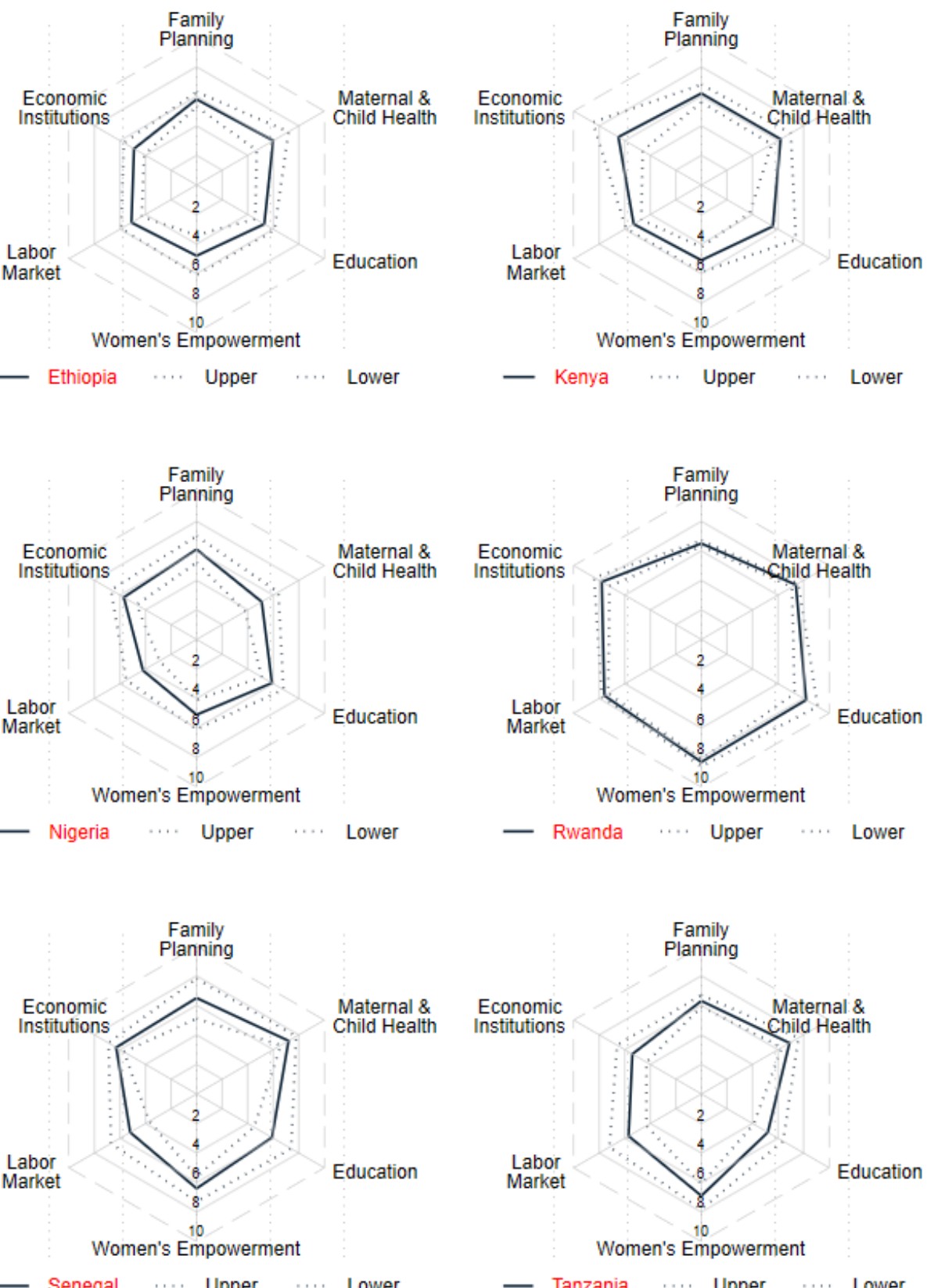

**Figure 1** Average scores per country and sector.

95% CI 2.6 to 5.3), advocacy (4.3, 95% CI 3.0 to 5.7) and research (4.4, 95% CI 3.0 to 5.8). The FP scores were highest in nearly all domains compared with the other sectors followed by ED. MCH scores consistently scored around the mid-point or lower in research (4.8, 95% CI 3.2 to 6.5) and civil society (4.6, 95% CI 2.8 to 6.5). The scores for WE were consistently second or third lowest across the domains.

In Rwanda, scores were mostly around 7 and none of the domains was scored below the mid-point. The scores were consistently higher for WE and ED across domains and the highest was in policymaking for WE (8.7, 95% CI 8.4 to 9.0) and research for ED (8.4, 95% CI 7.5 to 9.3). The lower scores were recorded in FP in nearly all the domains and the lowest score was in services and programmes (5.7, 95% CI 5.2 to 6.2). MCH and LM recorded second or third lowest scores across the five domains. GEI scored the third highest scores across all five domains.

In Senegal, most scores were above the mid-point across all sectors and domains except in LM. The scores were higher in MCH with the highest in services and programmes (8.0, 95% CI 7.3 to 8.7). The lowest scores were recorded across domains in LM, civil society (4.3, 95% CI 2.7 to 5.8), advocacy (4.5, 95% CI 2.8 to 6.2), research (4.7, 95% CI 3.2 to 6.3) and in services and programmes (4.8, 95% CI 3.4 to 6.3). The scores for WE were consistently around 6 or higher except in research (5.7, 95% CI 4.5 to 6.9). The scores for WE were consistently around 6 or higher, except in research (5.7, 95% CI 4.5 to 6.9).

In Tanzania, all the scores across sectors and domains were around 5 or higher. In nearly all the domains, WE scored higher with the highest score in civil society (8.1, 95% CI 7.5 to 8.6). The lowest scores were recorded mostly in ED and GEI with the lowest score in civil society participation in ED (5.0, 95% CI 3.3 to 6.6). MCH and FP scored second and third highest across domains, respectively, with the scores ranging from 6.2 (95% CI 5.5 to 6.9) to 6.9 (95% CI 6.1 to 7.6).

## Heterogeneity testing

The heterogeneity tests by domains across countries yielded similar results for fixed and random effect models. The $I^2$ estimates ranged from 0.0% to 55.5% in services and programmes for Rwanda. In all the other countries, $I^2$ was 0.0% across all the domains and sectors, suggesting no heterogeneity. $\tau^2$ was 0.0 across all the domains, sectors and countries, except in Rwanda where it was 0.46. For detailed heterogeneity testing results, see online supplemental appendix 5.

## DISCUSSION

This study aimed to develop a tool to assess national efforts in establishing a favourable policy environment to cultivate a DD. The study collected and analysed data across six key sectors in Nigeria, Rwanda, Ethiopia, Senegal, Tanzania and Kenya from key informants identified by the in-country study collaborators.

The DDEI scores revealed patterns that were helpful to guide discussions regarding needed improvements in respective countries across five domains: namely, policymaking, services and programmes, advocacy, research and civil society. The patterns of effort scores for the LM, WE, FP, GEI and ED sectors suggested great challenges for women's self-determination and empowerment, employment and leadership. These findings agreed with previous predictions by the World Economic Forum that women may have to wait for 100 years to pair men in leadership roles in bigger institutions and take up ministerial roles. However, in some countries, remarkable progress has been recorded indicating potential opportunity to attain gender parity much sooner. This is currently seen in the Global Gender Gap (GGG) Report, which ranks Rwanda in the top 10 countries with the smallest gender gap among countries worldwide.[38 39]

Relatedly, Rwanda scored remarkably high on WE, although the scores were also moderately high for Tanzania and Senegal. These findings aligned with how these countries performed on the 2022 GGG report, except for Kenya. Of the 146 countries ranked globally and on a maximum score possible of 1, Rwanda was ranked 6th with a score of 0.811 and Tanzania 64th with a score of 0.719. In contrast, Ethiopia ranked 74nd (0.710), Senegal 112th (0.668) and Nigeria 123th (0.639).[38] Kenya's rank has remarkably improved from the 2020 GGG report, having moved from 109th with a score of 0.671 to 57th with a score of 0.729. It is possible that the transformations that occurred in the last 2 years were not captured in the effort measurements, especially because the experts were asked about their experiences over the last 3–5 years prior to the DDEI survey. However, such rapid change in GGG rank indicates the importance of repeating the effort measurements on an annual basis. Furthermore, although the intent of the DDEI was not to rank countries, the study findings underscored the potential for mutual learning from countries and the utility of the DDEI to dive deeper into specific items that can reveal special lessons or areas that require improvements. This is especially crucial for countries that had a much larger change in ranking during the 2 years that preceded the report, such as Ethiopia which experienced an improvement of more than 40 ranks, and Kenya with a rise of 80 ranks in the last few years.[38 40]

Furthermore, human capital development is essential for a country to cultivate and harness its DD potential. The 2020 Human Capital Index report revealed that the scores for all six countries of this study ranged from 0.36 in Nigeria to 0.55 in Kenya. These findings indicate that children born today in the six countries could miss out on 45% to 64% of their future earnings potential because of incomplete ED and health that is not full.[41] The DDEI provided an opportunity to qualitatively compare different ongoing efforts initiated to develop human capital and the findings in MCH, ED and WE were in

agreement with the 2020 HCI findings. For instance, the lowest score for MCH was recorded in Nigeria (5.1, 95% CI 3.8 to 6.4), and domain-specific scores revealed larger gaps in research and civil engagement although the scores in the other domains were also low.

This study offers several strengths. First, the questionnaires and the implementation of the DDEI were conducted in collaboration with in-country partners. This collaboration fosters local ownership of the effort measurement tool. Second, high factor loadings and Cronbach's alpha suggested high internal consistency and reliability of the tool across countries. Our analysis also highlighted 38 items that did not seem to measure the same construct and were dropped from the analysis. Dropping these items improved the quality and length of the questionnaire of the DDEI. Third, the heterogeneity tests across domains revealed negligible heterogeneity in all countries except in the domain of services for Rwanda, which indicates score consistency across domains in these countries and the validity of the summary scores generated.[42] In the future, local institutions can regularly replicate these measurements at national and subnational levels. Fourth, unlike prior effort indices, such as the FPEI (created in 1972), the API (created in 1999) and MNPI (created in 2000),[27 28 43] the DDEI integrated critical sectors into one tool, and sectoral effort measures were not done in silos due to the interconnectedness and potential synergy between sectors (eg, FP, ED, MCH and WE). Finally, this tool provides information to constituents that could be used to hold their governments accountable and provide unconventional evidence for planning and evaluation purposes.

The DDEI was tested at the national level, which opened the opportunity to be applied at the subnational level. In countries where health and development programmes are decentralised, subnational DDEI can be transformational in guiding local budget allocation and self-evaluation. Given the paucity of quantitative data and poor utilisation of the existing information systems, the DDEI has the potential to become a unique data source that informs local discussions and conversations at all administrative and planning levels. However, special emphasis should always be made that beyond strategic allocation of resources, favourable changes in age structure accompanied by a mix of jobs that allow for optimal productivity for all workers are important preconditions to fully realise the country's DD potential.[44 45]

This study also presented a few limitations. The DDEI relied on experts' opinions, which may be affected by different factors. The respondent's judgement may be heavily influenced by their recent experience, which may not reflect the efforts placed over the years. Their judgement may be influenced by the level of their expectations, which may also be correlated with their level of satisfaction. For instance, where people have low expectations, the scores may be high despite the weak levels of effort and vice-versa. The challenges related to the reliability of observers' judgements have been documented

in other effort measurements such as the FPEI, API and MNPI.[28] However, despite this limitation, effort measures capture unique data that can guide planning and action, especially in countries where the availability of programme data is limited. Although the DDEI could be used as a monitoring tool to assess the progress in the indicators that have shown relevance or relationship to DD achievements in the literature, due to the duration that may be required for a DD to happen, the DDEI estimates should not be interpreted as a prediction of achieving a DD. In addition, in some of the countries, the sample size was small for some of the domains which may have affected the reliability and validity of tests. While the sample size could be limited by available knowledgeable respondents or experts in respective sectors for specific countries, the small sample size made it impossible to conduct confirmatory factor analysis at the country level, but was overcomed by pooling the data across countries.

Furthermore, other limitations worth mentioning are related to the nature of the responses and scores obtained from the respondents. The proportions of respondents who did not know the answer to the questions and consequently could not provide a score varied across sectors and countries, although no questions had a high (>10%) proportion of 'don't know' responses across all six countries. The proportions were notably high in MCH research in Kenya, Nigeria and Tanzania, in ED civil society in Kenya and Nigeria, in WE advocacy in Ethiopia, in LM research and civil society in Ethiopia, Kenya and Nigeria and GEI advocacy and civil society in Nigeria and Rwanda. This finding indicates that for future surveys, some of the questions could be amended to make 'don't know' items easier to interpret or removed where found ambiguous. For detailed percentages of 'don't know' responses received by item across countries, see online supplemental appendix 6. Another limitation is that the aggregate average scores for the sectors modulated centreing to the mid-point of 5 over the maximum possible 10. This tendency may indicate that respondents may have compensated for less generous scores that they gave to some questions with more generous scoring for others. However, this is less plausible because the item-specific scores within domains revealed wide variations across scores.

Lastly, the DDEI questionnaire was administered in the midst of the COVID-19 pandemic, which led to abrupt changes in health and development systems in all countries. Although respondents provided great insights about their respective sectors, it is not unrealistic to assume the response rates and responses received may have been affected by the status of pandemics in their countries. Additional research will be needed to nurture the index, include questions that may be used for validation and assess the need for weighing items and domains based on the stage of demographic transition, while improving the usability of the DDEI data by local and global health and development stakeholders.

## Conclusion

This study developed the DDEI for Nigeria, Ethiopia, Kenya, Tanzania, Rwanda and Senegal and demonstrated the utility of this index. The DDEI yielded several recommendations on opportunities to accelerate socio-economic development and take advantage of the DD potential in these countries. The scores for each sector and domain provided variations that allowed for comparison within a sector in each country. By engaging with local experts and partnering with local institutions in a participatory process, the data from the DDEI were used by local policymakers, service providers, programme implementers, advocates and researchers for self-evaluation and to develop recommendations to strengthen ongoing efforts and programmes in respective sectors.

The DDEI allowed for multisectoral analysis and provided an opportunity to identify sectors in most need and those that can serve as models in each country. For instance, based on sector-specific scores, LM and WE stood out as the sectors that required particular attention. However, despite higher sector-specific scores, a closer look at item-specific scores across domains revealed wide variations that were the basis of generating policy recommendations. The DDEI can be applied to all SSA countries regularly to monitor progress and foster local conversation around cultivating sustainable socio-economic development, which ultimately may contribute to realising the benefits of a DD.

The DDEI can serve as a complementary tool to existing global development monitoring tools. The findings from the DDEI aligned with other existing measures such as the GGG and HCI in various countries. Given the quantitative nature of these tools, the DDEI provides an opportunity to assess the inputs and guide policies, investments and programmes in specific sectors and items that can positively affect the gender gap and human capital. In addition to driving conversation among stakeholders in FP, MCH, ED, WE, GEI and LM, the DDEI focuses on assessing the inputs that are critical for the outcomes measured by GGG and HCI. The current DDEI has been translated into English and French. In the future, it should be translated into other languages such as Portuguese and Spanish to make it applicable to other countries in the early stages of their demographic transition.

**Acknowledgements** We express our deepest thanks to all participants for their contributions to this study. We would also like to thank our local partners and staff who facilitated this work at the Centre d'Excellence Régional en Economie Générationnelle in Senegal in particular Prof. Latif Dramani, Edem Akpo, Marthe Edmée Ndoye, Sam Agbahoungba, Mamaye Thionguane, Oumy Laye, Pierre Aloysius Ndiaye, Ndeye Dabakh, Malick Ndiaye and Boubacar Diallo. Our collaborators Charity Ibeawuchi, Mojisola Odeku, Olukunle Omotoso, Oluwayemisi Ishola and Olusola Obajimi from the Center for Communication Programs in Nigeria are highly appreciated. Our partners Mohamed A. Sheikh, Peter Nyakwara and Francis Kundu from the National Council for Population and Development in Kenya are also appreciated. We also appreciate the contribution made by the team from Tanzania Communication and Development Center especially Nazir Yusuph, Jacob Macha, Abubakar Msemo and Deo Ng'wanansabi. Our deepest thanks to the collaborators from Rwanda Kabano H. Ignace, Venant Habarugira, Rutayisire Pierre Claver, Benita Nyampundu, Dominique Kanobana, Gasafari Mpabuka Willy, Mazimpaka Jean Claude, Muhoza Dieudonne, Mbarushimana Nelson, Sebeza Jackson and Anicet Nzabonimpa. We also appreciate Assefa Admassie, Amdissa Teshome, Tamene Keneni Walga and Terefe Gelibo from the Ethiopian Academy of Sciences. All individuals listed here have been notified of their appearance in this manuscript.

**Contributors** JCR, CC, MM-B and JGR conceived the study. JCR, CC, MM-B, SA and JGR developed the questionnaire, led data collection, data analysis, and interpretation. JCR prepared the first draft of the manuscript. All authors revised and approved the final and revised versions of the paper and agree to be accountable for all aspects of the work. All authors accept full responsibility for the finished work, had access to the data, and controlled the decision to publish.

**Funding** This work was supported by the Bill and Melinda Gates Foundation under the grant OPP1181398.

**Competing interests** None declared.

**Patient and public involvement** Patients and/or the public were not involved in the design, or conduct, or reporting, or dissemination plans of this research.

**Patient consent for publication** Not applicable.

**Ethics approval** The study was conducted in compliance with the Helsinki declaration and was approved by the Johns Hopkins Bloomberg School of Public Health ethical review board (IRB Number: IRB00011522). All participants provided informed consent prior to taking part in this research.

**Provenance and peer review** Not commissioned; externally peer reviewed.

**Data availability statement** Data are available in a public, open access repository. The data and questionnaires used to generate this article are available on Harvard Dataverse open access repository accessible at https://doi.org/10.7910/DVN/ZSNPMV.

**ORCID iDs**
Jean Christophe Rusatira http://orcid.org/0000-0002-1538-0056
Carolina Cardona http://orcid.org/0000-0002-0570-001X

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
