## [Reviewer comments · BMJ Open]

ARTICLE DETAILS

TITLE (PROVISIONAL)	A new demographic dividend effort index (DDEI): results from an experts' survey from six sub-Saharan African countries.
AUTHORS	Rusatira, Jean Christophe; Cardona, Carolina; Martinez-Baack, Michelle; Rimon, Jose; Ahmed, Saifuddin

VERSION 1 – REVIEW

REVIEWER	Bloom, David Harvard School of Public Health, Department of Global Health and Population
REVIEW RETURNED	12-Apr-2022

GENERAL COMMENTS	This article describes the development, deployment, and results of a novel survey-based index designed to measure national efforts toward realizing the demographic dividend. The authors identify six thematic “pillars” (also referred to as “sectors”) on which the demographic dividend is necessarily built—good governance and economic institutions, family planning, maternal and child health, education, women’s empowerment, and labor market—and survey purposively sampled experts on their countries’ efforts to realize the demographic dividend via actions within their pillars of expertise. Survey questions are distributed across five dimensions of action—policy, services/programs, advocacy, research, and civil society—with experts drawn from each of these dimensions. Standardized on a 10-point scale, survey responses are averaged to compute scores for each pillar by country; overall country scores—the final outputs of the demographic dividend effort index, or DDEI—are obtained from an average of pillar scores. The authors deserve due credit for their efforts toward the creation of an eminently useful and timely tool. The demographic dividend indeed rests on many pillars, all of which must be deliberately and collaboratively erected by diverse stakeholders for the dividend to be realized. This intricacy calls for careful monitoring of stakeholders’ actions within each pillar and regular evaluation of progress using transparent, consistent metrics that allow stakeholders to hold one another accountable. The DDEI is an ambitious attempt to give stakeholders precisely this tool of monitoring and evaluation. With further calibration of the index and validation of its findings over successive waves of data collection, its potential to guide countries toward policies and partnerships conducive to realizing the demographic dividend will invariably grow. However, validation of the index may be a challenge; this is the subject of this review’s most substantial major comment. The literature on program effort indices—some of which the authors
---

cite—concedes that direct validation of such indices is impossible, as they rely on subjective judgments of largely qualitative variables. An indirect approach to validation consists of confirming a statistically significant relationship that is stable in time between index scores (at the country, pillar, dimension, or item level) and program outcomes. This approach is actionable if the effort index seeks to evaluate a program with outcomes that can be measured in a straightforward fashion (e.g., AIDS infections or deaths per capita, maternal or under-five deaths per thousand births, youth unemployment, etc.). However, there exists no such concisely quantifiable outcome when it comes to the demographic dividend. Even if there were a concisely quantifiable outcome of the demographic dividend, it would be of no present use to the authors: The DDEI is meant to collect data from countries that haven't yet started the demographic transition or are early in the transition; these countries are still years away from experiencing the demographic dividend (or not), so validation of the current results could only begin in the not-so-proximate future.

One (admittedly flawed) way to validate the data already gathered in this wave would consist of confirming significant relationships between sub-national index scores—that is, scores at the pillar, dimension, and item levels—and program outcomes that are specific to each pillar. These pillar-specific outcomes are effectively intermediate outputs which feed into the ultimate outcome (the demographic dividend), so their relationship with pillar-specific program effort could be measured over a nearer time horizon than that of the demographic dividend overall. However, this approach would not suffice to validate the DDEI as a whole, as the manner in which pillar scores are aggregated to compute country scores must itself be validated. A provisional means of overall validation could use preexisting indices with extensive historical coverage as shortcuts. For each of the pillars comprising the DDEI, it may be possible to find a thematically similar (“cousin”) index that recently scored the DDEI’s country sample and also previously scored countries which are now late in the demographic transition when these countries were pre-transition or early in the transition. These cousin indices may not all be program effort indices, though program effort indices are available for the pillars of family planning and maternal and child health; what matters most is the stability of their methodology between waves. For these cousin indices to serve as shortcuts to validation, the authors must show that (i) the DDEI pillar scores are robustly correlated with their cousin index scores, perhaps with controls for national development characteristics, and (ii) the cousin index scores were predictive of demographic dividend realization in countries that are now late in the demographic transition.

However, this roundabout approach to validation relies on the assumption that the ingredients of the demographic dividend (as proxied by the cousin indices) are the same today as they were when the countries now at the end of the demographic transition were at its beginning. Unless the authors can demonstrate this assumption’s validity, this shortcut to validation is necessarily provisional; validation must either wait until the DDEI’s sample countries have completed the demographic transition, or an alternative design for measuring national effort toward realization of the demographic dividend should be considered.

Perhaps such an alternative design would start with a rigorous modeling exercise of when in the demographic transition, and to what extent relative to other factors, the pillars used by the authors

feed into the realization of the demographic dividend. This exercise would necessarily rely on historical data from countries that have completed the demographic transition, and it would of course require the calibration of a model that identifies and measures the magnitude of any demographic dividend obtained. The modeling results could then indicate which programs outcomes within each pillar (e.g., fertility reduction, educational attainment, female labor force participation, etc.) are necessary to realize the demographic dividend, when in the demographic transition these outcomes are needed, and which interactions between outcomes have multiplicative effects on the magnitude of the demographic dividend. On this basis, the current outcomes of pre-transition and early transition countries could be benchmarked against the outcomes these countries would need to achieve in order to realize the demographic dividend. At this point, it would make sense to survey experts on various dimensions of program effort within each pillar in order to see where additional efforts are needed if countries are to meet their benchmarks. This research design makes surveying experts on program effort the last step rather than the first: Instead of using expert surveys to identify whether or not a country is off-course vis-à-vis realizing the demographic dividend, this task would be done by a thorough modeling exercise, after which expert surveys would give insights into pillar-specific areas that require additional effort to get a country back on track.

The other major comment concerns the manner in which individual respondents' item scores are aggregated and transformed into pillar and country scores. Nowhere do the authors delineate this process from start to finish, and this opens the door to misinterpretation of their methods. By the reviewer's calculations, the country scores in Table 2 more closely resemble a weighted average of each country's pillar scores, with the number of experts within each pillar used as weights, than an unweighted average of pillar scores. If this resemblance is a representation of fact, then the influence of each pillar score in a given country's overall score is proportional to the share of experts with expertise in that pillar who happened to respond to the survey from the given country. But the share of respondents with expertise in a given pillar varies massively by country: Among Senegalese respondents, only 8.6% reported expertise in family planning, versus 39.6% among Tanzanian respondents. It would therefore appear that family planning counts almost five times as heavily in Tanzania's country score than it does in Senegal's. In fairness, this apparent weighting system does not result in substantially different country scores than would have resulted from a simple average of pillar scores. Nonetheless, the authors must be transparent about how survey respondents' item scores were aggregated and averaged lest readers draw their own conclusions (as the reviewer has illustrated).

The following minor comments also warrant careful consideration by the authors:

- The authors should concretely list the outcome or outcomes that they identify with the demographic dividend. These outcomes are the dependent variables that would underpin the eventual validation of the DDEI, so their explicit enumeration is important. In parallel, the authors should discuss potential distributional consequences of the demographic dividend.
- The background section is missing a thorough treatment of the concept of a program effort index. Readers should be clear on its

	origins, evolution in practice, unique benefits relative to other methodologies, and suitability for the authors' enterprise. This treatment should also present generalizable critiques of the concept of a program effort index and explain how these concerns are addressed by the authors.  • Clarity is needed on why the six countries studied were selected. Are these countries representative of the spectrum of stages along the pre- and early transition? What elements of convenience informed this selection of countries, and which countries were ruled out on grounds of convenience? • A review of the program effort indices that inspired the DDEI make clear where the five dimensions (policy, services/programs, advocacy, research, civil society) of each of the six pillars come from. However, a comment on the logical framework that informed the selection of these five dimensions (and not the others that are represented in the literature) would be welcome. • A table of which items in the DDEI come from which preexisting program effort index questionnaires would be helpful. • Readers would be quite interested to know the details of the revisions undertaken to the questionnaire following the pilot test. On what criteria did pilot respondents evaluate the questionnaire? What analysis of pilot scores was performed, and what decisions were taken on this basis? • Respondents' organizational affiliations (public sector, private sector, NGO, university/research institute, or other) and areas of expertise (policy/policymaking, services or programs, advocacy, research, civil society) vary widely in their shares of each country's sample. Given the possibility of systematic differences in respondents' outlooks on the basis of affiliation and expertise, this source of potential incomparability across countries' scores should be explicitly addressed. • As the DDEI questionnaire is in its first revision, the authors should consider identifying and discussing survey items that appear problematic by virtue of a high "don't know" response rate, a high nonresponse rate, or high variance among respondents' scores. • Inasmuch as possible, the current text content of the results section should be presented in tables. Much of the text rehashes the content of Table 3; this doesn't seem entirely necessary. The text also summarizes the items measured in each dimension; a table could instead list these items more exactly. The text would be better spent highlighting and interpreting high-level trends (including the prevalence of the problematic items mentioned above). • The authors may want to consider presenting alternate country scores after application of a preliminary set of weights that adjust the importance of each pillar in light of each country's place in the demographic transition. These weights need not be final—only indicative of how future work could adjust the interpretation of the survey results. • Given the authors' concern about respondents' potential recency biases, they may wish to have respondents evaluate both the current situation and the situation as it existed two years prior (as the AIDS Program Effort Index does) or three years prior (as the Maternal and Neonatal Program Effort Index does) in future revisions of the DDEI. Doing so would not only mitigate recency bias, but also offer an ad hoc measure of sampling fluctuation if the DDEI is conducted at consistent two- or three-year intervals.
--	--

	 • The authors should justify their choice of a 10-point rating scale. Trials of the AIDS Program Effort Index suggest that respondents have difficulty distinguishing between adjacent increments of a 10-point rating scale and recommend a 5-point scale instead. • The results or discussion section would benefit from an analysis of covariance between scores on different items. This analysis could identify which stakeholders the respondents see as responsible for inadequate efforts in a certain area; it could also identify potentially redundant items. • A thorough copy-edit of the manuscript would be advisable in advance of resubmission. THIS REVIEW WAS PREPARED JOINTLY BY LEO ZUCKER AND DAVID BLOOM
--	--

REVIEWER	Sunde, Uwe Ludwig Maximilians University Munich, Economics
REVIEW RETURNED	19-Apr-2022

GENERAL COMMENTS	This paper presents the Demographic Dividend Effort Index, which is based on expert interviews about different sectors (governance and economic institutions, family planning, maternal and child health, education, women empowerment, and labor markets) in six countries in Sub-Saharan Africa (Ethiopia, Kenya, Nigeria, Rwanda, Senegal, and Tanzania). A set of expert respondents in each country answered a questionnaire comprising 490 items in total, spread over the six sectors. The study provides an overview of the questionnaire design, summarizes the sampling of respondents, and provides an overview of the results. Assessment The implementation of a Demographic Dividend Effort Index follows the example of other expert-based indexes (e.g. human resources effort index, family planning, etc.). The main limitations of the index are its comparability across respondents, countries and sectors, the lack of validation, and the quantitative interpretability. Here, few experts (<100) per country, with unclear background, respond to 490 items about six different topics (governance and economic institutions, family planning, maternal and child health, education, women empowerment, and labor markets) in six countries in Sub-Saharan Africa (Ethiopia, Kenya, Nigeria, Rwanda, Senegal, and Tanzania). While the attempt to get some objective information about the efforts of governments to foster the demographic development and the demographic dividend is commendable, the proposed methodology has many weaknesses that range from a lack of comparability of responses across individuals, items, and countries without prior data modifications up to potential problems of strategic replies. I am therefore skeptical about the validity and reliability of the results without validation of the index. 1. The index construction follows other expert indexes. The main concern is the interpretation of
--

	raw quantitative responses. Expert respondents provide information for 490 items, very often on the mid response category. It is well-known in survey design that responses to qualitative items are often hard to compare across individuals, at least without accounting for respondent-specific response behavior. Hence, the analysis of responses would require filtering systematic heterogeneity by respondents (respondent fixed effects, and systematic differences across experts with different positions, sexes, background) to obtain information with enhanced comparability. 2. The recruiting of experts is not fully transparent. A particular concern relates to strategic responses that are driven by political motives or “experimenter demand” considerations. Filtering responses by respondent background and analyzing the residual variation would be desirable in this context. 3. A validation of the index is lacking. For instance, one would expect that survey responses should match with policy activities in particular fields in a given country. For instance, this could be done with available statistical measures (such as, e.g., infant and maternal mortality in the context of maternal and child health, dynamics in female labor force participation rates in the context of women empowerment, etc.). No attempt of such a validation is made. 4. The analysis is restricted to comparisons of means and statistical significance of differences. Other ways of aggregating information (e.g., within sector or country) are conceivable. These include, for instance, data reduction by means of clustering of responses or factor analysis. This would also help making indexes comparable across sectors and domains. 5. A better use of the index instead of conducting cross-country comparison could be the analysis of repeated surveys within a country with the same experts. This would allow circumventing some of the concerns and deliver information about the dynamics in efforts to support the demographic dividend
--	--

VERSION 1 – AUTHOR RESPONSE

Reviewer: 1

Comment 1.

This article describes the development, deployment, and results of a novel survey-based index designed to measure national efforts toward realizing the demographic dividend. The authors identify six thematic “pillars” (also referred to as “sectors”) on which the demographic dividend is necessarily built—good governance and economic institutions, family planning, maternal and child health, education, women’s empowerment, and labor market—and survey purposively sampled experts on their countries’ efforts to realize the demographic dividend via actions within their pillars of expertise. Survey questions are distributed across five dimensions of action—policy, services/programs, advocacy, research, and civil society—with experts drawn from each of these dimensions. Standardized on a 10-point scale, survey responses are averaged to compute scores for each pillar

by country; overall country scores—the final outputs of the demographic dividend effort index, or DDEI—are obtained from an average of pillar scores.

The authors deserve due credit for their efforts toward the creation of an eminently useful and timely tool. The demographic dividend indeed rests on many pillars, all of which must be deliberately and collaboratively erected by diverse stakeholders for the dividend to be realized. This intricacy calls for careful monitoring of stakeholders' actions within each pillar and regular evaluation of progress using transparent, consistent metrics that allow stakeholders to hold one another accountable. The DDEI is an ambitious attempt to give stakeholders precisely this tool of monitoring and evaluation. With further calibration of the index and validation of its findings over successive waves of data collection, its potential to guide countries toward policies and partnerships conducive to realizing the demographic dividend will invariably grow.

However, validation of the index may be a challenge; this is the subject of this review's most substantial major comment. The literature on program effort indices—some of which the authors cite—concedes that direct validation of such indices is impossible, as they rely on subjective judgments of largely qualitative variables. An indirect approach to validation consists of confirming a statistically significant relationship that is stable in time between index scores (at the country, pillar, dimension, or item level) and program outcomes. This approach is actionable if the effort index seeks to evaluate a program with outcomes that can be measured in a straightforward fashion (e.g., AIDS infections or deaths per capita, maternal or under-five deaths per thousand births, youth unemployment, etc.). However, there exists no such concisely quantifiable outcome when it comes to the demographic dividend. Even if there were a concisely quantifiable outcome of the demographic dividend, it would be of no present use to the authors: The DDEI is meant to collect data from countries that haven't yet started the demographic transition or are early in the transition; these countries are still years away from experiencing the demographic dividend (or not), so validation of the current results could only begin in the not-so-proximate future. One (admittedly flawed) way to validate the data already gathered in this wave would consist of confirming significant relationships between sub-national index scores—that is, scores at the pillar, dimension, and item levels—and program outcomes that are specific to each pillar. These pillar-specific outcomes are effectively intermediate outputs which feed into the ultimate outcome (the demographic dividend), so their relationship with pillar-specific program effort could be measured over a nearer time horizon than that of the demographic dividend overall. However, this approach would not suffice to validate the DDEI as a whole, as the manner in which pillar scores are aggregated to compute country scores must itself be validated.

A provisional means of overall validation could use preexisting indices with extensive historical coverage as shortcuts. For each of the pillars comprising the DDEI, it may be possible to find a thematically similar (“cousin”) index that recently scored the DDEI's country sample and also previously scored countries which are now late in the demographic transition when these countries were pre-transition or early in the transition. These cousin indices may not all be program effort indices, though program effort indices are available for the pillars of family planning and maternal and child health; what matters most is the stability of their methodology between waves. For these cousin indices to serve as shortcuts to validation, the authors must show that (i) the DDEI pillar scores are robustly correlated with their cousin index scores, perhaps with controls for national development characteristics, and (ii) the cousin index scores were predictive of demographic dividend realization in countries that are now late in the demographic transition. However, this roundabout approach to validation relies on the assumption that the ingredients of the demographic dividend (as proxied by the cousin indices) are the same today as they were when the countries now at the end of the demographic transition were at its beginning. Unless the authors can demonstrate this assumption's validity, this shortcut to validation is necessarily provisional; validation must either wait until the

DDEI's sample countries have completed the demographic transition, or an alternative design for measuring national effort toward realization of the demographic dividend should be considered.

Perhaps such an alternative design would start with a rigorous modeling exercise of when in the demographic transition, and to what extent relative to other factors, the pillars used by the authors feed into the realization of the demographic dividend. This exercise would necessarily rely on historical data from countries that have completed the demographic transition, and it would of course require the calibration of a model that identifies and measures the magnitude of any demographic dividend obtained.

The modeling results could then indicate which programs outcomes within each pillar (e.g., fertility reduction, educational attainment, female labor force participation, etc.) are necessary to realize the demographic dividend, when in the demographic transition these outcomes are needed, and which interactions between outcomes have multiplicative effects on the magnitude of the demographic dividend. On this basis, the current outcomes of pre-transition and early transition countries could be benchmarked against the outcomes these countries would need to achieve in order to realize the demographic dividend. At this point, it would make sense to survey experts on various dimensions of program effort within each pillar in order to see where additional efforts are needed if countries are to meet their benchmarks. This research design makes surveying experts on program effort the last step rather than the first: Instead of using expert surveys to identify whether or not a country is off-course vis-à-vis realizing the demographic dividend, this task would be done by a thorough modeling exercise, after which expert surveys would give insights into pillar-specific areas that require additional effort to get a country back on track.

We thank the reviewer for providing excellent comments and suggestions.

Please note that the objective of the tool was to assess efforts in different health, social and developmental sectors that are currently underway or in place in a country to cultivate the demographic dividend. The tool is not attempting or designed to assess realized dividends from the support ratio or dependency ratio or the third stage of demographic transition. This is especially because the demographic transition offers policymakers a "window of opportunity" (Bloom, Canning et al. 2003) and the countries must utilize this window of opportunity in due time in order to realize a DD and avoid a "demographic disaster" (Canning, D., Raja, S., & Yazbeck, A. S. (Eds.). (2015). Africa's demographic transition: dividend or disaster?.(Canning, Raja et al. 2015) With this motivation, we have devised the tool to measure and track readiness so that the countries may take necessary steps towards addressing the gaps and lapses. Hence, "...validation must [either] wait until the DDEI's sample countries have completed the demographic transition" defeats the objective the DDEI tool. However, we have conducted extensive statistical reliability and validation testing of the tool to address the comments of the reviewers.

In response, we have also made corresponding changes in many sections and added a new section in the methods and results that describe the steps undertaken for statistical reliability and validity testing of the DDEI survey tool. We agree that validation is a challenge in the context of DD and using indirect method is not straightforward at this point. Given that there was no gold standard to which to compare the DDEI results for validation we took the following statistical steps to ensure the reliability and validity of the survey instrument:

Reliability testing:

We used Cronbach's alpha analysis to assess internal consistency of the tool and the results show that the items used in the questionnaire were highly intercorrelated within five domains across the six sectors, which implies that these items are highly linked to the measured latent constructs of the

sector-specific efforts. In the future as more surveys are conducted in the same countries, further tests for stability across waves will be conducted. We have also conducted statistical testing for heterogeneity testing to assess for consistency of scores by domains in each country.

Validity testing:

Given the unique nature of the DD as a concept, the temporality of efforts and when outcomes can be measured and the need for a multidimensional measurement, the validation relied on different forms of validation that were possible at the time of the study.

1. Content validity: Content validation was done through two approaches. First, the sectors and domains were identified based on an extensive literature review cited in the paper and sought feedback from experts who had field experience in different sectors included in the survey. Second, questionnaires of different sectors were developed based on the same review and integrated existing questionnaires where possible. These included the family planning effort index, the Family Planning Effort Index (FPEI), the Maternal and Neonatal Program Index (MNPI), the Rwanda Governance Scorecard, and the AIDS Program Effort Index (API).

2. Construct validity: Construct validity was done using confirmatory factor analysis and the results revealed reasonably high loadings and good model fitting based on the goodness-of-fit statistics.

To reflect these changes, we have added sections describing Cronbach's alpha, Confirmatory analysis and heterogeneity testing in the analysis and results sections. The revised texts are shown below in the respective sections for the reviewers' convenience:

Analysis

Cronbach's Alpha analysis

Because the data were collected separately by sector and country, Cronbach's alpha analysis was conducted by checking for "item-test" and "item-rest" correlations for each domain (five practice areas) for the six different sectors namely FP, MCH, ED, LM, WE and GGEI in each country. The item-rest correlations were used for examining the relevance of an item in the domain. The item-rest correlation indicated the association of an item with the total score on the other items, hence a score with items with higher item-rest correlations will result in a higher coefficient α for the test. (Lord and Novick 2008) To ensure the best performance of the score with the most relevant items, we decided to exclude the items only if they had an "item-rest" correlation of <0.4 across all countries. (De Groot and Van Naerssen 2018) This is because alpha values may vary when variables are context-sensitive and in the countries where the same variables indicate high reliability, the fraction of a test score that is attributable to error eventually decreases. Hence, despite the low item-rest correlation in some countries, items with high item-rest correlation in at least one country were used to measure the construct in question. (Henson 2001) The items that showed perfect pairwise correlation were excluded from the analysis.

Confirmatory factor analysis

Confirmatory factor analysis (CFA) was done using pooled data to allow for sufficient sample size using the "sem" Stata command that allows one item to be anchored with a value of 1.0 and the rest to have a loading value. The results were used to assess the fit of the retained data to measure the same underlying latent construct or domain. The expectation was that the loadings would not be too low, suggesting that the items we have included are appropriate and measuring the same constructs. Based on the results from this model, five indices were used to assess the goodness of fit of the data and whether the variables were used to measure the same construct. First, the Standardized Root Mean squared residual (SRMR) by which a good fit is indicated by a small value limited to 0.08. Second, the Coefficient of Determination (CD) by which, a model is said to have a good fit if it has a value close to 1. (Browne and Cudeck 1992) Third, the Root Mean Square Error of Approximation

(RMSEA), using its upper and lower bound; if the lower bound is below 0.05, the hypothesis that the fit is close is not rejected and if the upper bound is above 0.10, the hypothesis that the fit is poor is not rejected. (Bentler 1990) Fourth, Comparative Fit Index (CFI) by which a value close to 1 indicates a good fit. (Bentler 1990) Fifth, Tucker-Lewis index (TLI) by which a value close to 1 indicates a good fit. (Browne and Cudeck 1992)

Heterogeneity testing

Given that the measures were collected from different groups, we formally checked for heterogeneity statistically. Statistical testing for heterogeneity among respondents was done using meta-analytical techniques with the "metan" Stata command. This analysis used pooled data by country to allow for sufficient sample size and assessment of heterogeneity across domains. To assess the overall extent of heterogeneity and whether it was adequate to compute summary estimates, the fixed-effects were compared to random-effect estimates. By using this method, substantial discrepancies between the estimates from the two models indicate considerable heterogeneity that can make summary estimates misleading. (Poole and Greenland 1999) To further assess heterogeneity, the chi-square test of heterogeneity and I-square (I²) were used. The I² estimate of 0% to 40% indicate no/negligible heterogeneity, 30% to 60% indicate moderate heterogeneity, 50% to 90% indicate substantial heterogeneity and 75% to 100% show considerable heterogeneity. (Higgins, Thompson et al. 2003)

Results

Reliability and Validity

The alpha analysis revealed item-correlation estimates that were >0.4 in at least one country for all the items (variable), hence no items were excluded in further analysis. For detailed item-rest correlations for each item by domain, country and sector see Appendix 3. In the FP sector, the coefficients were >0.8 in all domains and countries except for the policy domain in Tanzania ($\alpha=0.72$). In the MCH sector, the coefficient estimates were 0.8 or higher in all other domains and countries except Senegal ($\alpha=0.59$). For the WE sector, the coefficient α estimates were >0.8 in all domains and countries. In the ED sector, the coefficient α estimates were >0.8 except for advocacy in Senegal ($\alpha=0.70$) and research in Nigeria ($\alpha=0.71$). For the labor market, the coefficient α estimates were >0.7 in nearly all countries and domains except advocacy in Ethiopia ($\alpha=0.40$). In the GGEI sector, the coefficient α estimates were >0.7 in all countries and domains.

The confirmatory analysis revealed reasonably high loadings and good model fitting: SRMR estimates ranged from 0.04 to 0.15 with the estimates for most domains across sectors estimated at around 0.08. The CD estimates ranged from 0.89 to 0.99 across domains and sectors. The RMSEA estimates range from 0.14-0.46 across domains and sectors. The lower bound ranged between 0.08-0.46 and the upper bound ranged between 0.16-0.59. The CFI estimates were around 0.6 or higher except for services and programs in FP and MCH. For detailed confirmatory analysis results by domains and sector see Appendix 4.

The heterogeneity tests by domains across countries yielded similar results for fixed and random effect models. The I² estimates ranged from 0.0% to 62.5% in Ethiopia. In Kenya, the I² estimate was highest in services and programs (22.9%) and lowest (0.0%) in Policy, research, and civil society. In Nigeria, I² estimates ranged from 22.4% in research to 45.1% in services and programs. In Rwanda, the estimates ranged from 76.6% in research to 94.6% in services and programs. In Senegal, the I² estimates ranged from 47.0% in research to 88.6% in advocacy. In Tanzania, the lowest estimate was in research (39.9%) and the highest was in civil society (80.2%). For detailed heterogeneity testing results see Appendix 5.

Further, as suggested by the reviewer, exploring relationships between sub-national index scores and program outcomes could be useful. However, the selection of specific outcome measures and availability of data across all sectors could be a challenge, hence we opted for a statistical validation of the content and the discriminative validation will be considered for future steps as more data points become available.

The suggestion by the review to search for “cousin” indices is highly appreciated. We also agree with the reviewer that the assumption about comparable/similar effect of the index score to the DD realization would require careful consideration. However, based on our comprehensive literature search, only the effort measures available were from the 2014 Family Planning Effort Index survey and no available data for similar indices for the other sectors considered for the DDEI. In addition, as widely documented, Sub-Saharan African countries have proven to behave differently in their pace of demographic transition and socio-economic growth compared to the other countries that were at the same levels of fertility in the 1970s which may be due to unique characteristics in these countries. (E. Bloom, Canning et al. 2007, Bongaarts 2015) This evidence makes developing models using data from other countries that have realized a demographic dividend such as those from Asia likely unrealistic for Africa. As documented by Bloom, without favorable age structure changes and mix of jobs that allow for optimal productivity of workers, countries cannot fully realize their DD potential. (Bongaarts and Casterline 2013) We have made edits to our discussion section to highlight this precondition to realizing a demographic dividend while avoiding misinterpretation of the DDEI scores.

The last paragraph of the discussion section now reads:

The DDEI was tested at the national level which opened the opportunity to be applied at the sub-national level. In countries where health and development programs are decentralized, sub-national DDEI can be transformational in guiding local budget allocation and self-evaluation. Given the paucity of quantitative data and poor utilization of the existing information systems, the DDEI has the potential to become a unique data source that informs local discussions and conversations at all administrative and planning levels. However, special emphasis should always be made that beyond strategic allocation of resources, favorable changes in age structure accompanied by a mix of jobs that allows for optimal productivity for all workers are important preconditions to fully realize the country’s DD potential. (Bloom, Canning et al. 2007, Bloom 2011)

The suggestion by the review for a modeling exercise is highly appreciated and could be a great addition to the study. However, this alternative design would require careful consideration of the inputs that may build from extensive work that has been done in the field as they relate to the sub-Saharan Africa countries. At this point, this alternative design is beyond the scope of the study but could be considered in future related work. We do not have sufficient data point across countries to explore the relationships of “population level outcomes” and the sector-specific effort indices but will consider doing it in the future as more data becomes available from other LMIC.

Further, the sample size is not adequate for a decomposition analysis, and it is almost impossible to know the relative significance of each action and their contribution to the DD either empirically or theoretically. Moreover, weighting by expertise level in domains in sector was not possible as response to expertise background was not mutually exclusive. So, we have given equal weight to all items in different domains and sectors in our analysis. However, the “total” score was weighted for differential response rates across the sectors (inverse of the response probability out of total targeted sample).

Comment 2

The other major comment concerns the manner in which individual respondents' item scores are aggregated and transformed into pillar and country scores. Nowhere do the authors delineate this process from start to finish, and this opens the door to misinterpretation of their methods. By the reviewer's calculations, the country scores in Table 2 more closely resemble a weighted average of each country's pillar scores, with the number of experts within each pillar used as weights, than an unweighted average of pillar scores. If this resemblance is a representation of fact, then the influence of each pillar score in a given country's overall score is proportional to the share of experts with expertise in that pillar who happened to respond to the survey from the given country. But the share of respondents with expertise in a given pillar varies massively by country: Among Senegalese respondents, only 8.6% reported expertise in family planning, versus 39.6% among Tanzanian respondents. It would therefore appear that family planning counts almost five times as heavily in Tanzania's country score than it does in Senegal's. In fairness, this apparent weighting system does not result in substantially different country scores than would have resulted from a simple average of pillar scores. Nonetheless, the authors must be transparent about how survey respondents' item scores were aggregated and averaged lest readers draw their own conclusions (as the reviewer has illustrated).

We thank the reviewer for providing this comment. We have revised our calculations for the overall score and made changes in the text to clarify that the overall score is a weighted average that allows each sector to have equal contribution to the overall score. The first paragraph of the analysis section has been edited to reflect these changes and now reads:

The characteristics of the participants are presented by sector and domain of expertise at the country level (Table 1). Then, mean scores are computed for ease of interpretation and comparison across sectors per country (Figure 1). Their corresponding 95% confidence intervals (CI) were calculated with a t-distribution because the sample size was relatively small. The weighted overall country scores were computed (Table 2) to allow for equal contribution of each sector to the overall score irrespective of the number of respondents in the sector and mean scores per sector were disaggregated by domain of expertise for ease of interpretability and policy relevance (Table 3). To test the reliability and validity of our estimates, Cronbach's alpha and confirmatory factor analyses were utilized. Heterogeneity testing was also done to assess the consistency of scores by domains in each country. Finally, the results from this analysis were shared with in-country partners, for review, policy translation and dissemination. The partners from respective countries conducted a series of multisectoral meetings and workshops, physical and virtual, to generate policy recommendations based on the results from the analysis. The data have been made available through the Harvard Dataverse open access repository. (Rusatira, Cardona et al. 2022)

Response to minor comments: The following minor comments also warrant careful consideration by the authors:

- The authors should concretely list the outcome or outcomes that they identify with the demographic dividend. These outcomes are the dependent variables that would underpin the eventual validation of the DDEI, so their explicit enumeration is important. In parallel, the authors should discuss potential distributional consequences of the demographic dividend.

We thank the reviewer for providing this comment. Our analysis did not attempt to identify any DD outcome(s) because the study's focus was on conceptual and practical processes that may create a favorable environment for the DD to happen. The potential distribution of consequences of DD could also not be generated from the effort measures collected.

- The background section is missing a thorough treatment of the concept of a program effort index. Readers should be clear on its origins, evolution in practice, unique benefits relative to other methodologies, and suitability for the authors' enterprise.

We thank the reviewer for providing this comment and we have made corresponding edits to provide more context on the origin, evolution, and rationale of effort measures. We have added a second last paragraph in the introduction which reads:

Over the last four decades, effort measurements have been proven to be reliable approaches to informing international health and development programs. Since the 1970s, the family planning effort index (FPEI) has provided reliable estimates to measure the strength of family planning programs in more than 90 countries and has been proven to be valid relative to family planning outcomes.(Mauldin, Berelson et al. 1978, Ross and Mauldin 1996, Ross and Stover 2000) Building on the success of the FPEI, the same approach has been applied in HIV/AIDS programs and Maternal and child health which led to the development of the AIDS Program Effort Index (API) in more than 35 countries (Stover 1999) and Maternal and Neonatal Program Index (MNPI) in more than 46 countries.(Ross, Campbell et al. 2001) However, because health and development systems do not operate in silos, the successful implementation of FPEI, API and MNPI present an opportunity to develop a multidimensional index to measure DD progress in SSA.

- This treatment should also present generalizable critiques of the concept of a program effort index and explain how these concerns are addressed by the authors.

We thank the reviewer for providing this comment and we have made corresponding edits on the first paragraph of the limitation section which now reads:

This study presented a number of limitations. The DDEI relied on experts' opinions, which may be affected by different factors. The respondent's judgment may be heavily influenced by their recent experience, which may not reflect the efforts placed over the years. Their judgment may be influenced by the level of their expectations, which may also be correlated with their level of satisfaction. For instance, where people have low expectations, the scores may be high despite the weak levels of effort, and vice-versa. The challenges related to the reliability of observers' judgments have been documented in other effort measurements such as the FPEI, API and MNPI.(Ross, Campbell et al. 2001) However, despite this limitation, effort measures capture unique data that can guide planning and action, especially in countries where the availability of program data is limited.

- Clarity is needed on why the six countries studied were selected. Are these countries representative of the spectrum of stages along the pre- and early transition? What elements of convenience informed this selection of countries, and which countries were ruled out on grounds of convenience?

We thank the reviewer for providing this comment and have made corresponding changes in the study design first paragraph which now reads:

Data were collected through an online survey that measured the efforts that countries are devoting to maximize their potential to reap the benefits of a DD. The survey was administered through an internet-based modality using the SurveyMonkey platform in six countries: Kenya, Senegal, Nigeria, Rwanda, Tanzania, and Ethiopia. These countries were selected because of where they stand in the demographic transition and convenience to successfully implement the study. The selection of countries sought to include both pre-dividend and early dividend countries(Bank 2015) in which the

study team had ongoing existing partners with the ability to engage a diverse pool of respondents. The survey(s) was administered in two waves to accommodate Coronavirus Disease 2019 (COVID-19) lockdowns in different countries. The first wave occurred from August to November 2020, and the second wave that covered Senegal alone took place from March to May 2021. The main language of the questionnaire was English (Appendix 1), and a French version (Appendix 2) was created for Senegal by a certified translator.

- A review of the program effort indices that inspired the DDEI make clear where the five dimensions (policy, services/programs, advocacy, research, civil society) of each of the six pillars come from. However, a comment on the logical framework that informed the selection of these five dimensions (and not the others that are represented in the literature) would be welcome.

We thank the reviewer for providing this comment and have made corresponding changes in the study design second paragraph which now reads:

The questionnaires were developed for each of the six sectors recommended by Cardona and colleagues: FP, MCH, ED, WE, LM, and GGEI.(Cardona, Rusatira et al. 2020) All questionnaires were structured around five domains: 1) policy; 2) services and programs; 3) advocacy; 4) research; and 5) civil society. These domains were selected based on prior similar effort indices implemented in various health and development areas(Stover 1999, Ross and Stover 2001, Ross, Campbell et al. 2001, Fort, Deussom et al. 2017). These domains also corresponded to the areas of practice in the field and were confirmed by the local partners during the study design process. The Demographic Dividend Effort Index (DDEI) questionnaire was composed of 490 items divided across the six sectors (112 items for FP, 101 items for MCH, 61 items for ED, 71 items for WE, 68 items for LM, 77 items for GGEI). Where available, the development of each sector-specific questionnaire integrated validated questions from existing effort indices. For instance, the Family Planning Effort Index (FPEI) questionnaire(Ross and Stover 2001) was integrated into the FP questionnaire. Similarly, specific questions from the Maternal and Neonatal Program Index (MNPI)(Ross, Campbell et al. 2001), the Rwanda Governance Scorecard(Usengumukiza, Munyandamutsa et al. 2017) and the AIDS Program Effort Index (API)(Stover 1999) were integrated into different sector-specific questionnaires. Additional questions were drafted based on DD-favorable interventions for respective sectors as summarized in the wheel of prosperity by Cardona et al.(Cardona, Rusatira et al. 2020) During the data collection, participants only ever received single surveys that pertained to their sector of expertise and never the entirety of the aggregate survey. Each item was scored on a scale of 1 to 10, from weaker (1) to stronger (10) perceived level of effort or strength.

- A table of which items in the DDEI come from which preexisting program effort index questionnaires would be helpful.

We appreciate this comment from the reviewer. The study questionnaire was provided as an appendix and can be cross-checked with respective effort indices. Due to time and space constraints the table that matches each question to respective of effort questionnaires could not be produced at this point.

- Readers would be quite interested to know the details of the revisions undertaken to the questionnaire following the pilot test. On what criteria did pilot respondents evaluate the questionnaire? What analysis of pilot scores was performed, and what decisions were taken on this basis?

We thank the reviewer for providing this comment and have made corresponding changes in the study design third paragraph which now reads:

The DDEI questionnaire was pilot-tested among 27 experts from Nigeria, Ethiopia, Tanzania, Senegal, Kenya and Rwanda in June-August 2020. This pilot study allowed local experts to provide their feedback and inputs to the questionnaire, which were integrated into the final revision. The local experts were requested to provide qualitative feedback on content, conciseness and completeness by sector and were requested to provide recommendations on changes to be made. The final questionnaire was shared with local project partners for review and approval before administering to the national level across all six countries. Each sector-specific questionnaire included items that assessed the presence and strength of policies, services and programs, advocacy efforts, research, and the involvement of civil society organizations in that sector. No personal identification or socio-demographic characteristics information was collected to limit the concern of being identifiable for the respondents and minimize information bias.

- Respondents' organizational affiliations (public sector, private sector, NGO, university/research institute, or other) and areas of expertise (policy/policymaking, services or programs, advocacy, research, civil society) vary widely in their shares of each country's sample. Given the possibility of systematic differences in respondents' outlooks on the basis of affiliation and expertise, this source of potential incomparability across countries' scores should be explicitly addressed.

We thank the reviewer for providing this comment. Due to the limited sample size an analysis to explore systematic differences by respondents' affiliation would not provide reliable estimates. An important consideration is that the pool of "experts" is also limited and the sample size possible vary by country. We acknowledge this challenge and have statistically tested for heterogeneity and included a section on heterogeneity testing in the methodology and result sections. We have also acknowledged this limitation in the first two paragraphs of the limitation section that read:

This study presented a number of limitations. The DDEI relied on experts' opinions, which may be affected by different factors. The respondent's judgment may be heavily influenced by their recent experience, which may not reflect the efforts placed over the years. Their judgment may be influenced by the level of their expectations, which may also be correlated with their level of satisfaction. For instance, where people have low expectations, the scores may be high despite the weak levels of effort, and vice-versa. The challenges related to the reliability of observers' judgments have been documented in other effort measurements such as the FPEI, API and MNPI. (Ross, Campbell et al. 2001) However, despite this limitation, effort measures capture unique data that can guide planning and action, especially in countries where the availability of program data is limited.

Furthermore, some of the items maintained in the score had low item-rest correlation estimates in some of the countries although they had higher estimates in other countries. This may suggest that some items may be context dependent. For multi-country applications of this tool, these items were not dropped because their item-rest correlations were low in one setting, but not in others. In some of the countries, the sample size was small for some of the domains which may have affected the reliability and validity tests. In Rwanda, Senegal and Tanzania, heterogeneity estimates were high. Although heterogeneity may be due to chance alone, these findings warrant a larger sample size in future similar surveys. Nevertheless, the sample size could be limited by available knowledgeable respondents or experts in respective sectors for specific countries.

- As the DDEI questionnaire is in its first revision, the authors should consider identifying and discussing survey items that appear problematic by virtue of a high “don’t know” response rate, a high nonresponse rate, or high variance among respondents’ scores.

We thank the reviewer for providing this comment . We have generated an additional appendix (Appendix 6) that highlights percentages of “don’t know” responses received by item/question across countries. We have also made changes to add corresponding details in second last paragraph of the limitation section that now reads:

Further, other limitations worth mentioning are related to the nature of the responses and scores obtained from the respondents. The proportions of respondents who did not know the answer to the questions and consequently could not provide a score varied across sectors and countries although no questions had a high (>10%) proportion of “don’t know” across all the six countries. The proportions were notably high in MCH research in Kenya, Nigeria and Tanzania, in ED civil society in Kenya and Nigeria, in WE advocacy in Ethiopia, in LM research and civil society in Ethiopia, Kenya and Nigeria and in GGEI advocacy and civil society in Nigeria and Rwanda. This finding indicates that for future surveys, some of the questions could be amended to make “don’t know” answers easier to interpret or removed where found ambiguous. For detailed percentages of “don’t know” responses received by item across countries see Appendix 6. Another limitation is that the aggregate average scores for the sectors modulated centering to the mid-point of 5 over the maximum possible 10. This tendency may indicate that respondents may have compensated for less generous scores that they gave to some questions with more generous scoring for others. However, this is less plausible because the item-specific scores within domains revealed wide variations across scores.

- In as much as possible, the current text content of the results section should be presented in tables. Much of the text rehashes the content of Table 3; this doesn’t seem entirely necessary. The text also summarizes the items measured in each dimension; a table could instead list these items more exactly. The text would be better spent highlighting and interpreting high-level trends (including the prevalence of the problematic items mentioned above).

We thank the reviewer for providing this comment. Because of the amount of information collected across six sectors, the results section was structured to allow for the readers to get a sense of where the scores came from and highlights the key findings in each sector and domain. This is because different readers may be interested in different aspects of the DDEI, and the study approach was to allow for equal representation of different sectors. The text highlighting and interpreting the key findings was included in the discussion section.

- The authors may want to consider presenting alternate country scores after application of a preliminary set of weights that adjust the importance of each pillar in light of each country’s place in the demographic transition. These weights need not be final—only indicative of how future work could adjust the interpretation of the survey results.

We thank the reviewer for providing this comment. However, given the fact that DD would not have specific explicit outcomes for the study countries – the estimation of relative weights of different sectors could be unrealistic at this point. This study focused on measuring the necessary steps to create a favorable environmental for the country to take advantage of their window of DD opportunity. In the seminal DD literature by Bloom and colleagues, it has been acknowledged that creating a favorable policy environment will be key for developing countries to take advantage of their demographic dividend potential. The same literature, which was also cited in the paper, highlights the diversity across nations that may drive differential response to similar interventions or policies. This

complexity makes assigning levels of importance to different sectors unrealistic if the same weights would be applied to different countries.

- Given the authors' concern about respondents' potential recency biases, they may wish to have respondents evaluate both the current situation and the situation as it existed two years prior (as the AIDS Program Effort Index does) or three years prior (as the Maternal and Neonatal Program Effort Index does) in future revisions of the DDEI. Doing so would not only mitigate recency bias, but also offer an ad hoc measure of sampling fluctuation if the DDEI is conducted at consistent two- or three-year intervals.

We thank the reviewer for providing this comment. The study team had considered measuring situations as they were three years before the survey in reference to the AIDS Program Effort Index (API) but did not find sufficient justification for a three-years period across DDEI sectors and countries. For the majority of sectors and countries, strategic plans and major investments are reviewed every 5-10 years which implied that for the majority of countries the trend that could be captured in a three-year period could be misleading depending on when the country reviewed or implemented their strategic/ development plans.

- The authors should justify their choice of a 10-point rating scale. Trials of the AIDS Program Effort Index suggest that respondents have difficulty distinguishing between adjacent increments of a 10-point rating scale and recommend a 5-point scale instead.

We thank the reviewer for providing this comment. The choice of 5 vs a 10-range scale was one of the subjects discussed during the questionnaire review with the local partners. The 10-point scale was recommended because it offers a wider range/variability, and this scale was more commonly used across study countries. During the consultations, it was also mentioned that the majority of countries' education systems tend to score from 0-10 which indicated that the respondents were more used to this scoring scale.

- The results or discussion section would benefit from an analysis of covariance between scores on different items. This analysis could identify which stakeholders the respondents see as responsible for inadequate efforts in a certain area; it could also identify potentially redundant items.

We thank the reviewer for providing this comment and have made corresponding changes in the article by adding alpha analysis and confirmatory analysis. The item-rest correlation from alpha analysis indicated the association of an item with the total score on the other items, which also indicate that where a score has items with higher item-rest correlations the Cronbach's α will be higher. The confirmatory factor analysis was used to assess the fit of the retained data to measuring the same underlying latent construct or domain. This analysis also allows to identify any items with poor loadings.

- A thorough copy-edit of the manuscript would be advisable in advance of resubmission.

We thank the reviewer for providing this comment. We have made sure the article is copy edited before submission.

Reviewer 2

Comment 1: The index construction follows other expert indexes. The main concern is the interpretation of raw quantitative responses. Expert respondents provide information for 490 items, very often on the mid response category. It is well-known in survey design that responses to qualitative items are often hard to compare across individuals, at least without accounting for

respondent-specific response behavior. Hence, the analysis of responses would require filtering systematic heterogeneity by respondents (respondent fixed effects, and systematic differences across experts with different positions, sexes, background) to obtain information with enhanced comparability.

We thank the reviewer for providing this comment and have made corresponding changes by making clarification edits on the study design section and adding heterogeneity test results.

The second paragraph in the study design section now reads:

The questionnaires were developed for each of the six sectors recommended by Cardona and colleagues: FP, MCH, ED, WE, LM, GGEI.(Cardona, Rusatira et al. 2020) All questionnaires were structured around five domains: 1) policy; 2) services and programs; 3) advocacy; 4) research; and 5) civil society. These domains were selected based on prior similar effort indices implemented in various health and development areas(Stover 1999, Ross and Stover 2001, Ross, Campbell et al. 2001, Fort, Deussom et al. 2017). These domains also corresponded to the areas of practice in the field and were confirmed by the local partners during the study design process. The Demographic Dividend Effort Index (DDEI) questionnaire was composed of 490 items divided across the six sectors (112 items for FP, 101 items for MCH, 61 items for ED, 71 items for WE, 68 items for LM, 77 items for GGEI). Where available, the development of each sector-specific questionnaire integrated validated questions from existing effort indices. For instance, the Family Planning Effort Index (FPEI) questionnaire(Ross and Stover 2001) was integrated in the FP questionnaire. Similarly, specific questions from the Maternal and Neonatal Program Index (MNPI)(Ross, Campbell et al. 2001), the Rwanda Governance Scorecard(Usengumukiza, Munyandamutsa et al. 2017) and the AIDS Program Effort Index (API)(Stover 1999) were integrated in different sector-specific questionnaires. Additional questions were drafted based on DD-favorable interventions for respective sectors as summarized in the wheel of prosperity by Cardona et al.(Cardona, Rusatira et al. 2020) During the data collection, participants only ever received single surveys that pertained to their sector of expertise and never the entirety of the aggregate survey. Each item was scored on a scale of 1 to 10, from weaker (1) to stronger (10) perceived level of effort or strength.

To address the issue of possible systematic heterogeneity by respondents, we used meta-analytical technique to examine the extent of heterogeneity across the domains in each country. An additional section has been added to the paper in both the method and result sections. We have also made the edits in the discussion and strength and limitation sections to reflect the heterogeneity testing results.

Lastly, no data were collected for cadre, length of work experience, sex and other background characteristics of the respondents, and the sample sizes would not have been sufficient to check domain specific variability by the respondent characteristics. We discussed this as the study limitation in Discussion section of the manuscript.

Comment 2: The recruiting of experts is not fully transparent. A particular concern relates to strategic responses that are driven by political motives or “experimenter demand” considerations. Filtering responses by respondent background and analyzing the residual variation would be desirable in this context.

We agree with this comment and have amended the text in study participant section. We have expanded the description of the selection process, mode of contacting and call back procedures. To minimize experimenter demand, the survey was self-administered, and no personal identification information were collected. However, we cannot filter the scores by respondent background to

analyze the residual variations by respondent characteristics because these characteristics were not collected to limit the fear to be identifiable for the respondents which could have biased our findings.

The first two paragraphs in the study participant section now read:

The in-country partners identified the potential participants who were knowledgeable in respective domains and sectors in each country. In-country partners included the Centre d'Excellence Régional en Economie Générationnelle in Senegal, the Center for Communication Programs in Nigeria, the National Council for Population and Development in Kenya, the Tanzania Communication and Development Center, the Ethiopian Academy of Sciences, and the Centre for Integrated Development Research and Action in Rwanda. Participants were considered for the survey if they were regarded to have demonstrated expertise in at least one of the six sectors by the in-country partner. Expertise was defined by having worked in one of the health or development sectors at the national level as a policymaker, advocate, service provider, researcher, or member of a civil society organization. The local study partners had the best knowledge of potential respondents and led the data collection process locally.

As with other similar indices, sampling was necessarily purposive given the need to identify knowledgeable respondents. The design entailed the selection of 10 respondents for each sector. Whenever possible, diversity across the five domains of expertise was prioritized. As such, to reach 10 respondents per sector, each of the six sectors were designed to have two respondents with sector expertise in one of the five domains. These were exclusive classifications such that respondents could only be counted towards fulfillment of one sector-specific questionnaire. Overall, the sample size goal was 60 respondents per country. The survey was administered through an online modality "SurveyMonkey" in each country. Local partners in respective countries appointed a contact person to respond to any queries regarding the study. Requests to fill the surveys were sent through existing national technical working groups email lists and up to three follow-up calls were made to all individuals for improving survey response rates.

Comment 3: A validation of the index is lacking. For instance, one would expect that survey responses should match with policy activities in particular fields in a given country. For instance, this could be done with available statistical measures (such as, e.g., infant and maternal mortality in the context of maternal and child health, dynamics in female labor force participation rates in the context of women empowerment, etc.). No attempt of such a validation is made.

We thank the reviewers for providing this comment and have made corresponding changes. We have added a new section on construct reliability and validity testing using Cronbach's alpha analysis and confirmatory factor analysis. In the future, we will conduct discriminant validation with a larger sample. Because the objective was not to identify subdomain, we have not attempted to reduce the item domains based on characteristics roots or eigenvalues.

The analysis section now reads:

Analysis

The characteristics of the participants are presented by sector and domain of expertise at the country level (Table 1). Then, mean scores are computed for ease of interpretation and comparison across sectors per country (Figure 1). Their corresponding 95% confidence intervals (CI) were calculated with a t-distribution because the sample size was relatively small. The weighted overall country scores were computed (Table 2) to allow for equal contribution of each sector to the overall score irrespective

of the number of respondents in the sector and mean scores per sector were disaggregated by domain of expertise for ease of interpretability and policy relevance (Table 3). To test the reliability and validity of our estimates, Cronbach's alpha and confirmatory factor analyses were utilized. Heterogeneity testing was also done to assess the consistency of scores by domains in each country. Finally, the results from this analysis were shared with in-country partners, for review, policy translation and dissemination. The partners from respective countries conducted a series of multisectoral meetings and workshops, physical and virtual, to generate policy recommendations based on the results from the analysis.

Cronbach's Alpha analysis

Because the data was collected separately by sector and country, Cronbach's alpha analysis was conducted with checking for "item-test" and "item-rest" correlations for each domain (five practice areas) for the six different sectors namely FP, MCH, ED, LM, WE and GGEI in each country. The item-rest correlations were used for examining the relevance of an item in the domain. The item-rest correlation indicated the association of an item with the total score on the other items, hence a score with items with higher item-rest correlations will result in a higher coefficient α for the test. (Lord and Novick 2008) To ensure the best performance of the score with most relevant items, we decided to exclude the items only if they had "item-rest" correlation of <0.4 across all countries. (De Groot and Van Naerssen 2018) This is because alpha values may vary when variables are context-sensitive and in the countries where the same variables indicate high reliability, the fraction of a test score that is attributable to error eventually decreases. Hence, despite the low item-rest correlation in some countries, items with high item-rest correlation in at least one country was used to measure the construct in question. (Henson 2001) The items that showed perfect pairwise correlation were excluded from the analysis.

Confirmatory factor analysis

Confirmatory factor analysis was done using pooled data to allow for sufficient sample size using the "sem" Stata command that allows one item to be anchored with value 1.0 and the rest to have a loading value. The results were used to assess the fit of the retained data to measuring the same underlying latent construct or domain. The expectation was that the loadings would not be too low, suggesting that the items we have included are appropriate and measuring the same constructs. Based on the results from this model, five indices were used to assess the goodness of fit of the data and whether the variables used measure the same construct. First, the Standardized Root Mean squared residual (SRMR) by which a good fit is indicated by a small value limited to 0.08. Second, the Coefficient of Determination (CD) by which, a model is said to have a good fit if it has a value close to 1. (Browne and Cudeck 1992) Third, the Root Mean Square Error of Approximation (RMSEA), using its upper and lower bound; if the lower bound is below 0.05, the hypothesis that the fit is close is not rejected and if the upper bound is above 0.10, the hypothesis that the fit is poor is not rejected. (Bentler 1990) Fourth, Comparative Fit Index (CFI) by which a value close to 1 indicates a good fit. (Bentler 1990) Fifth, Tucker-Lewis index (TLI) by which a value close to 1 indicates a good fit. (Browne and Cudeck 1992)

Heterogeneity testing

Given that the measures were collected from different groups, we formally checked for heterogeneity statistically. Statistical testing for heterogeneity among respondents was done using meta-analytical techniques with the "metan" Stata command. This analysis used pooled data by country to allow for sufficient sample size and assessment of heterogeneity across domains. To assess the overall extent of heterogeneity and whether it was adequate to compute summary estimates, the fixed-effects were compared to random-effect estimates. By using this method, substantial discrepancies between the estimates from the two models indicate considerable heterogeneity that can make summary estimates misleading. (Poole and Greenland 1999) To further assess heterogeneity, chi-square test of

heterogeneity and I-square (I2) were used. The I2 estimate of 0% to 40% indicate no/negligible heterogeneity, 30% to 60% indicate moderate heterogeneity, 50% to 90% indicate substantial heterogeneity and 75% to 100% show considerable heterogeneity.(Higgins, Thompson et al. 2003)

An additional section has been added to the results that reads:

Reliability and Validity

The alpha analysis revealed item-correlation estimates that were >0.4 in at least one country for all the items (variable), hence no items were excluded in further analysis. For detailed item-rest correlations for each item by domain, country and sector see Appendix 3. In the FP sector, the coefficients were >0.8 in all domains and countries except for the policy domain in Tanzania ($\alpha=0.72$). In the MCH sector, the coefficient estimates were 0.8 or higher in all other domains and countries except Senegal ($\alpha=0.59$). For the WE sector, the coefficient α estimates were >0.8 in all domains and countries. In the ED sector, the coefficient α estimates were >0.8 except for advocacy in Senegal ($\alpha=0.70$) and research in Nigeria ($\alpha=0.71$). For the labor market, the coefficient α estimates were >0.7 in nearly all countries and domains except advocacy in Ethiopia ($\alpha=0.40$). In the GGEI sector, the coefficient α estimates were >0.7 in all countries and domains.

The confirmatory analysis revealed reasonably high loadings and good model fitting: SRMR estimates ranged from 0.04 to 0.15 with the estimates for most domains across sectors estimated around 0.08. The CD estimates ranged from 0.89 to 0.99 in across domains and sectors. The RMSEA estimates ranging from 0.14-0.46 across domains and sectors. The lower bound ranged between 0.08-0.46 and the upper bound ranged between 0.16-0.59. The CFI estimates were around 0.6 or higher except for services and programs in FP and MCH. For detailed confirmatory analysis results by domains and sector see Appendix 4.

The heterogeneity tests by domains across countries yielded similar results for fixed and random effect models. The I2 estimates ranged from 0.0% to 62.5% in Ethiopia. In Kenya, I2 estimate was highest in services and programs (22.9%) and lowest (0.0%) in Policy, research, and civil society. In Nigeria, I2 estimates ranged from 22.4% in research to 45.1% in services and programs. In Rwanda, the estimates ranged from 76.6% in research to 94.6% in services and programs. In Senegal, the I2 estimates ranged from 47.0% in research to 88.6% in advocacy. In Tanzania, the lowest estimate was in research (39.9%) and highest in civil society (80.2%). For detailed heterogeneity testing results see Appendix 5.

We have expanded the discussion of the strengths and limitations of the study. The fifth paragraph in the discussion section now reads:

This study offers several strengths. First, the questionnaires and the implementation of the DDEI were conducted in collaboration with in-country partners. This collaboration fosters local ownership of the effort measurement tool. Second, high Cronbach's alpha and high loading of the items in CFA tool suggested high reliability and validity of the tool across countries. These tests also highlighted some items that appeared to be least reliable in selected countries; this information may be used to reduce the items of questionnaires in respective countries. The heterogeneity tests across domains revealed negligible to moderate heterogeneity in Ethiopia, Kenya and Nigeria, which indicate score consistency across domains in these countries and validity of the summary scores generated.(Fletcher 2007) In the future, local institutions can regularly replicate these measurements at national and sub-national levels. Third, unlike prior effort indices, such as the FPEI, (created in 1972), the API (created in 1999), and MNPI (created in 2000), (Lapham and Mauldin 1985, Stover 1999, Ross, Campbell et al. 2001) the DDEI integrated critical sectors into one tool, and sectoral effort measures were not done in silos

due to the interconnectedness and potential synergy between sectors (e.g. family planning, education, maternal and child health and women’s empowerment). Finally, this tool provides the information to constituents to hold their governments accountable and provide unconventional evidence for planning and evaluation purposes.

The first paragraph in the limitation section now reads:

This study presented a number of limitations. The DDEI relied on experts’ opinions, which may be affected by different factors. The respondent’s judgment may be heavily influenced by their recent experience, which may not reflect the efforts placed over the years. Their judgment may be influenced by the level of their expectations, which may also be correlated with the level of satisfaction. For instance, where people have low expectations, the scores may be high despite the weak levels of efforts, and vice-versa. The challenges related to the reliability of observers’ judgements have been documented in other effort measurements such as the FPEI, API and MNPI.(Ross, Campbell et al. 2001) However, despite this limitation, effort measures capture unique data that can guide planning and action especially in countries where availability of program data is limited.

Comment 4: The analysis is restricted to comparisons of means and statistical significance of differences. Other ways of aggregating information (e.g., within sector or country) are conceivable. These include, for instance, data reduction by means of clustering of responses or factor analysis. This would also help making indexes comparable across sectors and domains.

While we agree with the comment, in the paper we describe that the sectors and domains were already predefined before the analysis and were used in structuring the questionnaire and data collection process. Hence the data reduction methods would not be applicable in this context. These domains were identified based on literature and consultation with local partners working in health and development sectors. In this regard, the goal of the study was not to identify the latent constructs underlying each sector because the domains/components within each sector were already defined.

Comment 5: A better use of the index instead of conducting cross-country comparison could be the analysis of repeated surveys within a country with the same experts. This would allow circumventing some of the concerns and deliver information about the dynamics in efforts to support the demographic dividend.

The study team agrees. Repeated surveys within countries with the same experts will be highly valuable. This is future plan for better utilization of the DDEI.

VERSION 2 – REVIEW

REVIEWER	Bloom, David Harvard School of Public Health, Department of Global Health and Population
REVIEW RETURNED	04-Aug-2022
GENERAL COMMENTS	The reviewer appreciates the authors’ careful attention to the comments they provided upon reviewing the initial version of the manuscript. In response to some of these comments, the authors made revisions that reasonably address the reviewer’s concerns. However, the reviewer regards efforts to address their major comments as incomplete.

	The first major comment raised in the previous review challenged the authors to offer proof of the external validity of their index—in other words, that index scores are associated with achievement of desired program outcomes. At the time, the reviewer conceded that the validity of a program effort index is notoriously difficult to prove, at least over a short time horizon. Perhaps understandably, the authors did not include proof of their index’s external validity among their revisions. While external validation can remain an item of future work, the authors must not insinuate that their survey instrument and the results of its initial administration offer a robust basis for prediction of countries’ experiences with the demographic dividend. Assertions that the index is valid and reliable should be duly qualified. Perhaps in lieu of proof of external validation, the authors performed a battery of internal consistency tests on the scores they obtained from the index’s initial administration. Broadly speaking, these tests aim to confirm whether respondents’ scores on a group of items all measure the same latent concept which is captured by the aggregation of these items into a composite score. Discussion of these tests constitutes the most substantial change to the manuscript, and their use and presentation warrants a few comments. Five of these tests fall under the header of confirmatory factor analysis. As factor analysis appears in the literature as a means of evaluating the components of program effort indices, its use by the authors seems appropriate. However, the results of these five tests often fail to meet the thresholds of internal consistency that the authors mention when introducing the tests. For instance, the authors state that a Root Mean Squared Error of Approximation (RMSEA) requires a lower bound value below 0.05 to fail to reject the null hypothesis of good factor choice and an upper bound value below 0.1 to reject the hypothesis of poor factor choice; yet, per the values presented in Appendix 4, these thresholds are not attained for any domain under any sector. Similar remarks can be made of the other four confirmatory factor analyses to a lesser extent: None of the results presented in Appendix 4 lead the reviewer to the ineluctable conclusion that the authors’ desired threshold values or optimal values have been attained or approached, respectively. Yet the principal reference to these test results in the main text is an assertion that the results indicate good model fit, followed by a few value ranges of the results in extreme brevity. Since their assertion of good model fit is undermined by scrutiny of Appendix 4, the authors need to find some other way to reinforce their assertion with additional proof or otherwise qualify it with forthright interpretation of the suboptimal results they obtained from the tests employed in their revision. The authors also test respondents’ scores on each item using Cronbach’s alpha. Similar in motivation to confirmatory factor analysis, this test rewards high covariance among item scores within a domain as evidence that the items all represent the latent variable that is the aggregate score over the domain. As such, the authors choice to perform a Cronbach’s alpha analysis is defensible. However, the authors’ threshold for excluding a questionnaire item from score aggregation—alpha below 0.4 in all countries—seems exceptionally low. Henson (2001, pg. 181), whom the authors cite, reviews the literature on internal consistency tests and relates a variety of recommended thresholds for coefficient alpha; the lowest of these thresholds is 0.5, though significantly higher values were the standard in research at the
--	---

time of writing. Incidentally, the authors' chosen threshold is sufficiently low that none of the 490 items in their questionnaire appears to have qualified for exclusion. This outcome is attributable to not only the low threshold for alpha, but also the considerable variability of a single question's alpha across countries. Both factors merit further discussion in the main text. Last, the authors test respondents' scores for heterogeneity—that is, whether similar respondents gave significantly different scores to the same items. The reviewer is unable to interpret the results of these tests. The authors appear to group respondents by country and domain. If appearances are borne out in fact, this grouping is a curious one because respondents answered questions relevant to all five domains, not only their own, but only within their single sector of expertise. The authors thus seem to be testing whether respondents from the same country and from similar professional backgrounds scored similarly the entirely different sectors in which they work. The apparent underlying assumption of the test is, for example, that two Kenyan researchers—one of whom studies maternal and child health, the other of whom studies the labor market—should similarly score effort levels in their respective sectors of expertise by virtue of their both being Kenyan researchers. As the reviewer finds assumptions like this minimally credible, the authors must clarify why they aggregate respondents as they do when testing for heterogeneity. They must also explain why the two tables in Appendix 5 are identical in all their shared variables in spite of the different estimation methods (and respondent groupings) mentioned in their titles.

The second major comment raised in the previous review instructed the authors to explicitly state how they aggregate respondents' scores on individual items at the domain, sector, and country levels. The reviewer made this injunction after identifying what appeared to be an unstated weighting system underlying the aggregate values reported in Table 2. The authors' revised manuscript now states that the overall country scores at the bottom of Table 2 are weighted, but the weighting system allows "for equal contribution of each sector to the overall score irrespective of the number of respondents in the sector". If the reviewer has understood correctly, the authors' weighting system at the country level is that of assigning each sector score a weight of one—the definition of an unweighted average. Yet, by the reviewer's calculations, the overall country scores are still more plausibly produced by an average of sector scores with weights proportional to the number of respondents per sector than by an unweighted average of sector scores, and this even when rounding errors among the sector scores are taken into account. The manuscript's discussion of score aggregation therefore remains inadequate.

The authors addressed some of the minor comments provided in the initial review while electing not to address others:

- The authors do not delineate how they would identify the magnitude of a demographic dividend, which would be necessary for external validation of their index. They also do not discuss distributional consequences of the demographic dividend.
- The authors offer some additional background regarding the concept of the program effort index, its uses, and its limitations, although further elaboration would be welcome.
- The authors succinctly and satisfactorily clarify their reasons for selecting the six countries in which their questionnaire was initially administered.

	 • While the manuscript does not explain why the authors chose the index's five domains at the expense of others that appear in program effort indices, it now offers sufficient justification of these domains' validity. • The authors do not specify which items in their index are lifted from other program effort indices. • The manuscript now contains slightly more information about the feedback received on the questionnaire during the pilot test, though additional information on how this feedback resulted in changes to the questionnaire would be beneficial. • The authors do not elaborate on how cross-country variations in the distribution of respondents across domains may affect the comparability of country-level scores, though interpretation of the authors' internal consistency analyses could potentially assuage concerns in this area. • The authors responded thoroughly to the comment regarding problematic questionnaire items by presenting and analyzing the share of "don't know" responses by item and also by testing responses for heterogeneity (although the structure of the heterogeneity tests is confusing, as stated above). If nonresponse to certain items was frequent among respondents who completed the questionnaire, an analysis of these trends would be helpful too. • The authors minimally restructured the results section. The reviewer maintains that this section is not clearly organized. Given the difficulties in making cross-country score comparisons, the authors may wish to structure the results section as six discrete country profiles that focus on unique national strengths and weaknesses (the current results section is disproportionately centered on the latter). Such a structure may be more relevant and engaging to stakeholders. • The authors decline to compute alternate scores using provisional weights that reflect each country's progress in the demographic transition. This is understandable as producing even provisional weights would entail a large upfront investment of author time. • The authors do not explain their choice of a 10-point rating scale as opposed to a five-point scale, whose superiority has been suggested by tests of other program effort indices. This is a minor detail, however. • By performing and discussing internal consistency tests on respondents' scores, the authors now analyze item covariance for purposes of identifying extraneous and redundant items (though it appears that none were found). • A further copy-edit of the manuscript, particularly of newly added text, is necessary.
--	---

VERSION 2 – AUTHOR RESPONSE

Reviewer: 1

Dr. David Bloom, Harvard School of Public Health

Comments to the Author:

The reviewer appreciates the authors' careful attention to the comments they provided upon reviewing the initial version of the manuscript. In response to some of these comments, the authors made revisions that reasonably address the reviewer's concerns. However, the reviewer regards efforts to address their major comments as incomplete.

The first major comment raised in the previous review challenged the authors to offer proof of the external validity of their index—in other words, that index scores are associated with achievement of desired program outcomes. At the time, the reviewer conceded that the validity of a program effort index is notoriously difficult to prove, at least over a short time horizon. Perhaps understandably, the authors did not include proof of their index's external validity among their revisions. While external validation can remain an item of future work, the authors must not insinuate that their survey instrument and the results of its initial administration offer a robust basis for prediction of countries' experiences with the demographic dividend. Assertions that the index is valid and reliable should be duly qualified.

We thank the reviewer for providing this comment and suggestion. While describing the limitations of the study in the Discussion section, we have emphasized the point that the tool was not devised to predict the DD achievement based on the domain indicators or scores, but as a monitoring tool to assess the progress in the indicators that have shown relevance or relationship to DD achievements in the literature.

Perhaps in lieu of proof of external validation, the authors performed a battery of internal consistency tests on the scores they obtained from the index's initial administration. Broadly speaking, these tests aim to confirm whether respondents' scores on a group of items all measure the same latent concept which is captured by the aggregation of these items into a composite score. Discussion of these tests constitutes the most substantial change to the manuscript, and their use and presentation warrants a few comments.

Five of these tests fall under the header of confirmatory factor analysis. As factor analysis appears in the literature as a means of evaluating the components of program effort indices, its use by the authors seems appropriate. However, the results of these five tests often fail to meet the thresholds of internal consistency that the authors mention when introducing the tests. For instance, the authors state that a Root Mean Squared Error of Approximation (RMSEA) requires a lower bound value below 0.05 to fail to reject the null hypothesis of good factor choice and an upper bound value below 0.1 to reject the hypothesis of poor factor choice; yet, per the values presented in Appendix 4, these thresholds are not attained for any domain under any sector. Similar remarks can be made of the other four confirmatory factor analyses to a lesser extent: None of the results presented in Appendix 4 lead the reviewer to the ineluctable conclusion that the authors' desired threshold values or optimal values have been attained or approached, respectively. Yet the principal reference to these test results in the main text is an assertion that the results indicate good model fit, followed by a few value ranges of the results in extreme brevity. Since their assertion of good model fit is undermined by scrutiny of Appendix 4, the authors need to find some other way to reinforce their assertion with

additional proof or otherwise qualify it with forthright interpretation of the suboptimal results they obtained from the tests employed in their revision.

We thank the reviewer for providing great comments and suggestions.

We agree that our results previously provided in Appendix 4 were not close to the threshold described in the paper. Because of sample size limitations, structural equational modeling was omitted from the new iteration of the manuscript and corresponding appendix 4 was omitted. As suggested by Hu and Bentler (1999), structural equation models tend to overreject true-population models at small sample size and thus are less preferable when sample size is small.(Hu and Bentler 1999) Structural equational modeling will be a subject for future analysis once more data is available.

In our revised analysis, we used factor analysis results to identify items that needed to be dropped from the survey in order to refine the tool. We have improved the reliability and construct validity by using a factor loading cutoff of 0.5. A new section has been added to the paper describing factor and Cronbach's alpha analyses.

Factor and Cronbach's alpha analysis

Factor analysis was implemented to assess construct internal validity for each domain in each sector. This process allowed us to identify the latent structure revealed by the data. The analysis was conducted by respective sectors retaining one factor for each domain to identify which items loaded on that domain. This process allowed us to assess how each of the items included in the questionnaire loaded on the domain under which the item was administered. Items were retained if their factor loadings were 0.5 or higher. This threshold has been recommended by Costello and Osborne as a more reliable threshold that allows for the selection of items that strongly influence the domain, which improves the internal validity of the measurement.(Costello and Osborne 2005) Cronbach's alpha estimates were generated to assess the reliability of the retained items for each domain. A coefficient above 0.60 signals an adequate reliability coefficient.(DeVellis 2012)

A corresponding section has also been added to the results section.

Factor and Cronbach's alpha analysis

The items that loaded poorly (<0.5) were dropped from further analysis, which reduced the number of items from 112 to 85 items for FP, 101 to 94 items for MCH, 68 to 65 items for LM and 77 to 76 items for GEI. For detailed factor analysis results by domain and sector, see Appendix 4. Cronbach's alpha analysis results revealed excellent reliability with alpha estimates ranging from 0.84 in LM to 0.98 in ED (Table 2).

Table 2. Results from Cronbach's alpha analysis

Sector	Domain				
	Policy Civil Society	Services and Programs	Advocacy	Research	
Family Planning	0.85	0.93	0.93	0.95	0.97
Maternal and Child Health	0.94	0.99	0.92	0.98	0.97
Education	0.95	0.92	0.89	0.97	0.98
Women's Empowerment	0.98	0.96	0.94	0.98	0.97
Labor Market	0.95	0.95	0.84	0.96	0.97
Governance and Economic Institutions	0.95	0.97	0.94	0.97	0.94

This analysis is limited by the sample size. To achieve a number of experts that would be required to obtain more reliable results from a confirmatory factor analysis, the survey would have to cover more countries because the pool of experts in respective sectors is limited. Hence a confirmatory factor analysis will be the subject of future analysis once data from more countries is available.

The authors also test respondents' scores on each item using Cronbach's alpha. Similar in motivation to confirmatory factor analysis, this test rewards high covariance among item scores within a domain as evidence that the items all represent the latent variable that is the aggregate score over the domain. As such, the authors' choice to perform a Cronbach's alpha analysis is defensible. However, the authors' threshold for excluding a questionnaire item from score aggregation—alpha below 0.4 in all countries—seems exceptionally low. Henson (2001, pg. 181), whom the authors cite, reviews the literature on internal consistency tests and relates a variety of recommended thresholds for coefficient alpha; the lowest of these thresholds is 0.5, though significantly higher values were the standard in research at the time of writing. Incidentally, the authors' chosen threshold is sufficiently low that none of the 490 items in their questionnaire appears to have qualified for exclusion. This outcome is attributable to not only the low threshold for alpha, but also the considerable variability of a single question's alpha across countries. Both factors merit further discussion in the main text.

We thank the reviewer for this comment. We have revised our analysis and made corresponding edits in the discussion section.

We agree that the threshold of 0.4 was low and have revised our analysis accordingly. Due to the sample size limitation to estimate Cronbach's alpha for each item, we have instead used factor analysis to identify redundant items to be excluded from the analysis. The overall Cronbach's alpha for each domain was estimated and showed excellent reliability as they range from 0.84 to 0.98. We have elaborated more on this in the fifth and seventh paragraphs in the discussion section (p.11):

“This study offers several strengths. First, the questionnaires and the implementation of the DDEI were conducted in collaboration with in-country partners. This collaboration fosters local ownership of the effort measurement tool. Second, high factor loadings and Cronbach’s alpha suggested high internal consistency and reliability of the tool across countries. Our analysis also highlighted 38 items that did not seem to measure the same construct and were dropped from the analysis. Dropping these items improved the quality and length of the questionnaire of the DDEI. Third, the heterogeneity tests across domains revealed negligible heterogeneity in all countries except in the domain of services for Rwanda, which indicates score consistency across domains in these countries and the validity of the summary scores generated.(Fletcher 2007) In the future, local institutions can regularly replicate these measurements at national and sub-national levels. Fourth, unlike prior effort indices, such as the FPEI, (created in 1972), the API (created in 1999), and MNPI (created in 2000),(Lapham and Mauldin 1985, Stover 1999, Ross, Campbell et al. 2001) the DDEI integrated critical sectors into one tool, and sectoral effort measures were not done in silos due to the interconnectedness and potential synergy between sectors (e.g. family planning, education, maternal and child health, and women’s empowerment). Finally, this tool provides information to constituents that could be used to hold their governments accountable and provide unconventional evidence for planning and evaluation purposes.

....

This study also presented a few limitations. The DDEI relied on experts’ opinions, which may be affected by different factors. The respondent’s judgment may be heavily influenced by their recent experience, which may not reflect the efforts placed over the years. Their judgment may be influenced by the level of their expectations, which may also be correlated with their level of satisfaction. For instance, where people have low expectations, the scores may be high despite the weak levels of effort, and vice-versa. The challenges related to the reliability of observers’ judgments have been documented in other effort measurements such as the FPEI, API and MNPI.(Ross, Campbell et al. 2001) However, despite this limitation, effort measures capture unique data that can guide planning and action, especially in countries where the availability of program data is limited. Although the DDEI could be used as a monitoring tool to assess the progress in the indicators that have shown relevance or relationship to DD achievements in the literature, due to the duration that may be required for a DD to happen, the DDEI estimates should not be interpreted as a prediction of achieving a DD. In addition, in some of the countries, the sample size was small for some of the domains which may have affected the reliability and validity of tests. While the sample size could be limited by available knowledgeable respondents or experts in respective sectors for specific countries, the small sample size made it impossible to conduct reliable confirmatory factor analysis.”

Last, the authors test respondents’ scores for heterogeneity—that is, whether similar respondents gave significantly different scores to the same items. The reviewer is unable to interpret the results of these tests. The authors appear to group respondents by country and domain. If appearances are borne out in fact, this grouping is a curious one because respondents answered questions relevant to all five domains, not only their own, but only within their single sector of expertise. The authors thus seem to be testing whether respondents from the same country and from similar professional backgrounds scored similarly the entirely different sectors in which they work. The apparent underlying assumption of the test is, for example, that two Kenyan researchers—one of whom studies maternal and child health, the other of whom studies the labor market—should similarly score effort levels in their respective sectors of expertise by virtue of their both being Kenyan researchers. As the

reviewer finds assumptions like this minimally credible, the authors must clarify why they aggregate respondents as they do when testing for heterogeneity. They must also explain why the two tables in Appendix 5 are identical in all their shared variables in spite of the different estimation methods (and respondent groupings) mentioned in their titles.

We thank the reviewer for providing this comment.

Testing for heterogeneity was motivated by the fact that respondents' organizational affiliations (public sector, private sector, NGO, university/research institute, or other) and areas of expertise (policy/policymaking, services or programs, advocacy, research, civil society) varied widely in their shares of each country's sample. We tested for heterogeneity grouping ratings by domains to assess for heterogeneity within domains. This assessment was useful in giving us an idea of how homogenous the ratings of a specific domain in a specific sector in each country were. The ratings would have been concerning if they appeared highly heterogenous in specific domains given the fact that these are the areas of practice that inform experts' ratings. Alternatively, we could have explored systematic differences by respondents' affiliation, but the limited sample size would not allow us to obtain reliable estimates. An important consideration is that the pool of "experts" is also limited and the sample size possible varies by country.

On why the two tables in appendix 5, now appendix 6 in the new paper version, compare the results from fixed-effect and random-effect models. Because the models were implemented by domain, large discrepancies between the results would indicate high random effect by the domain. In other words, if there were large discrepancies between the fixed-effect and random-effect models, it would indicate between-domain heterogeneity which would mean that the fixed-effect assumption is incorrect. In this case, confirming that the different experts who scored similar domains in different sectors did not provide similar scores.

We have made changes to the description of heterogeneity testing in the analysis section (p.7), which now reads:

Heterogeneity testing

Given that the measures were collected from different groups, we formally checked for heterogeneity statistically. Statistical testing for heterogeneity among respondents was done using meta-analytical techniques. This analysis used pooled data by country to allow for sufficient sample size and assessment of heterogeneity across domains. To assess the overall extent of heterogeneity and whether it was adequate to compute summary estimates, the fixed-effects were compared to random-effect estimates. By using this method, substantial discrepancies between the estimates from the two models indicate considerable heterogeneity that can make summary estimates misleading. (Poole and Greenland 1999) To further assess heterogeneity across domains in respective countries I-squared (I²) and Tau-squared were used. The I² values of 25%, 50%, and 75% have been interpreted as representing small, moderate, and high levels of heterogeneity. (Higgins, Thompson et al. 2003) Tau-squared captures the variance between measurements of efforts by domains which can be captured by random-effect meta-analytic models.

and to the result section (p.10), which reads:

Heterogeneity testing

The heterogeneity tests by domains across countries yielded similar results for fixed and random effect models. The I2 estimates ranged from 0.0% to 55.5% in services and programs for Rwanda. In all the other countries I2 was 0.0% across all the domains and sectors, suggesting no heterogeneity. Tau-squared was 0.0 across all the domains, sectors, and countries, except in Rwanda where it was 0.46. For detailed heterogeneity testing results, see Appendix 5.

The second major comment raised in the previous review instructed the authors to explicitly state how they aggregate respondents' scores on individual items at the domain, sector, and country levels. The reviewer made this injunction after identifying what appeared to be an unstated weighting system underlying the aggregate values reported in Table 2. The authors' revised manuscript now states that the overall country scores at the bottom of Table 2 are weighted, but the weighting system allows "for equal contribution of each sector to the overall score irrespective of the number of respondents in the sector". If the reviewer has understood correctly, the authors' weighting system at the country level is that of assigning each sector score a weight of one—the definition of an unweighted average. Yet, by the reviewer's calculations, the overall country scores are still more plausibly produced by an average of sector scores with weights proportional to the number of respondents per sector than by an unweighted average of sector scores, and this even when rounding errors among the sector scores are taken into account. The manuscript's discussion of score aggregation therefore remains inadequate.

We thank the reviewer for providing this comment and have made corresponding changes to the article. We agree the language used in the previous version was confusing and have made changes accordingly.

A revised paragraph on how the index scores were generated has been added to the analysis section:

Index scores

The items retained from factor analysis were used to compute mean scores for countries, sectors, and domains. Sectoral and domain scores were computed as the simple average across the items retained. The total score for each country was computed as a weighted average that accounts for differences in the number of participants across sectors. Hence, sectoral weights for the total score were constructed as the ratio between the number of participants within a sector and the total number of participants for each country. For all scores, corresponding 95% confidence intervals (CI) were

calculated with a t-distribution because the sample size was relatively small. These estimates were presented graphically for ease of interpretation and comparison across sectors per country.

The results section has also been revised accordingly. The first paragraph under index scores results now reads:

The weighted overall country scores were computed to account for differences in the number of participants per sector. For example, in Ethiopia, there were 17 participants from the family planning sector, while there were only 8 participants in the women's empowerment sector. It would not be a balanced score if it assigned the same weight across sectors. Overall, the weighted average scores ranged from 5.4 (95% CI:5.1-5.8) in Ethiopia to 7.7 (95% CI:7.5-8.0) in Rwanda (Table 3). The second highest scores were for Tanzania (6.3, 95% CI:6.0-6.6) and Senegal (6.3, 95% CI:5.9-6.7). The second and third lowest scores were for Nigeria (5.5, 95% CI: 5.1-5.9) and Kenya (5.9, 95% CI: 5.5-6.2), respectively.

The authors addressed some of the minor comments provided in the initial review while electing not to address others:

- The authors do not delineate how they would identify the magnitude of a demographic dividend, which would be necessary for external validation of their index. They also do not discuss distributional consequences of the demographic dividend.

We thank the reviewer for this comment and suggestion. The current work focused on conceptualizing the tool and has focused on ensuring content and internal validity and reliability of the tool. Future work will include collecting more data to allow for domain-specific and overall external validation of the tool.

- The authors offer some additional background regarding the concept of the program effort index, its uses, and its limitations, although further elaboration would be welcome.

We thank the reviewer for this comment. We have reviewed the program effort index literature in each domain and listed key references for interested readers.

- The authors succinctly and satisfactorily clarify their reasons for selecting the six countries in which their questionnaire was initially administered.
- While the manuscript does not explain why the authors chose the index's five domains at the expense of others that appear in program effort indices, it now offers a sufficient justification of these domains' validity.
- The authors do not specify which items in their index are lifted from other program effort indices.

We thank the reviewer for this comment and suggestion. We agree to specify which items came from which index will be helpful. We have created a detailed table showing which items were lifted from different indices. This has been provided as Appendix 3.

- The manuscript now contains slightly more information about the feedback received on the questionnaire during the pilot test, though additional information on how this feedback resulted in changes to the questionnaire would be beneficial.

We thank the reviewer for this comment. We have provided additional information on the pilot and review process in the study design section, in the last paragraph, which now reads:

The DDEI questionnaire was pilot-tested among 27 experts from Nigeria, Ethiopia, Tanzania, Senegal, Kenya and Rwanda from June to August 2020. This pilot study allowed local experts to provide their feedback and inputs to the questionnaire, which were integrated into the final revision. The local experts were requested to provide qualitative feedback on content, conciseness, and completeness by sector, and were requested to provide recommendations on changes to be made. Following the feedback from local experts, the language and content of the questionnaire were amended to improve clarity and conciseness. The final questionnaire was shared with local project partners for final review and approval before administering at the national level across all six countries. Each sector-specific questionnaire included items that assessed the presence and strength of policies, services and programs, advocacy efforts, research, and the involvement of civil society organizations in that sector. No personal identification or socio-demographic characteristics information was collected to limit the concern of being identifiable for the respondents and minimize information bias.

- The authors do not elaborate on how cross-country variations in the distribution of respondents across domains may affect the comparability of country-level scores, though interpretation of the authors' internal consistency analyses could potentially assuage concerns in this area.
- The authors responded thoroughly to the comment regarding problematic questionnaire items by presenting and analyzing the share of "don't know" responses by item and also by testing responses for heterogeneity (although the structure of the heterogeneity tests is confusing, as stated above). If nonresponse to certain items was frequent among respondents who completed the questionnaire, an analysis of these trends would be helpful too.

We thank the reviewer for this comment. The main points about the nonresponses have been provided in the main text of the manuscript and more extensive details were provided in appendix 6. Interested readers will be able to identify the questions with higher non-response rates in each study country.

- The authors minimally restructured the results section. The reviewer maintains that this section is not clearly organized. Given the difficulties in making cross-country score comparisons, the authors may wish to structure the results section as six discrete country profiles that focus on unique national

strengths and weaknesses (the current results section is disproportionately centered on the latter). Such a structure may be more relevant and engaging to stakeholders.

We thank the reviewer for the comment and agree with the suggestion. We have restructured the results section and highlighted key results by country. The last six paragraphs of the results section now read:

“In Ethiopia, across the six sectors, none scored 5 or higher across all five domains. The scores were lowest for civil society engagement relative to other domains in FP, MCH, LM, and GEI. In most cases, policy or policymaking scored highest across sectors. Of all the scores, the highest average score was recorded in FP advocacy (6.9, 95% CI:6.0-7.9), and the lowest in research for WE (3.8, 95% CI:2.0-5.6) as well as civil society engagement in LM (3.8, 95% CI:2.3-5.4). The other scores that were below the mid-point were for services and programs in WE (4.7, 95% CI:3.3-6.1) and LM (4.8, 95% CI:4.1-5.4), advocacy in LM (4.7, 95% CI:4.1-5.4), research in WE (3.8, 95% CI:2.0-5.6) and GEI (4.6, 95% CI:3.6-5.6), and civil society engagement in MCH (4.9, 95% CI:3.4-6.4), ED (4.0, 95% CI:3.1-4.9) and GEI 4.7 (3.5-5.8). However, in most instances, the differences in scores between domains were not statistically significant due to overlapping confidence intervals.

In Kenya, the average scores were higher than the mid-point across sectors and domains except for research in LM (4.6, 95% CI:3.7-5.4) and civil society in MCH (4.9, 95% CI:3.2-6.7). Scores were around 6 for FP in all the domains. Similarly, except for civil society, the scores were around 6 for MCH. The scores for ED, WE and LM were mostly around 5 and hardly reached 6. GEI scored consistently highest compared to the other sectors across the five domains with the highest score in advocacy (7.0, 95% CI:4.5-9.5).

In Nigeria, most scores reached the mid-point or higher except for LM which hardly reached 5 in any of the five domains. LM scored lowest in services and programs (3.7, 95% CI:2.4-5.1), followed by policymaking (3.9, 95% CI:2.6-5.3), advocacy (4.3, 95% CI:3.0-5.7) and research (4.4, 95% CI:3.0-5.8). The FP scores were highest in nearly all domains compared to the other sectors followed by ED. MCH scores consistently scored around the mid-point or lower in research (4.8, 95% CI:3.2-6.5) and civil society (4.6, 95% CI:2.8-6.5). The scores for WE were consistently second or third lowest across the domains.

In Rwanda, scores were mostly around 7 and none of the domains were scored below the mid-point. The scores were consistently higher for WE and ED across domains and the highest was in policymaking for WE (8.7, 95% CI:8.4-9.0) and research for ED (8.4, 95% CI:7.5-9.3). The lower scores were recorded in FP in nearly all the domains and the lowest score was in services and programs (5.7, 95% CI:5.2-6.2). MCH and LM recorded second or third-lowest scores across the five domains. GEI scored the third highest scores across all five domains.

In Senegal, most scores were above the mid-point across all sectors and domains except in LM. The scores were higher in MCH with the highest in services and programs (8.0, 95% CI:7.3-8.7). The

lowest scores recorded were in civil society roles in LM (4.3, 95% CI:2.7-5.8), advocacy (4.5, 95% CI:2.8-6.2), research and in services and programs (4.8, 95% CI:3.4-6.3). The scores for WE were consistently around 6 or higher, except in research (5.7, 95% CI:4.5-6.9).

In Tanzania, all the scores across sectors and domains were around 5 or higher. In nearly all the domains, WE scored higher with the highest score in civil society (8.1, 95% CI:7.5-8.6). The lowest scores were recorded mostly in ED and GEI with the lowest score in civil society participation in ED (5.0, 95% CI:3.3-6.6). MCH and FP scored second and third highest across domains, respectively, with the scores ranging from 6.2 (95% CI:5.5-6.9) to 6.9 (95% CI:6.1-7.6)."

- The authors decline to compute alternate scores using provisional weights that reflect each country's progress in the demographic transition. This is understandable as producing even provisional weights would entail a large upfront investment of author time.
- The authors do not explain their choice of a 10-point rating scale as opposed to a five-point scale, whose superiority has been suggested by tests of other program effort indices. This is a minor detail, however.

We thank the reviewer for the comment. An explanation has been provided in the last two sentences of the second paragraph of the study design section that read:

Each item was scored on a scale of 1 to 10, from weaker (1) to stronger (10) perceived level of effort or strength. The scoring range of 1-10 has been used in FPEI and API(Stover 1999, Ross and Stover 2001) and allowed a wide variability needed to detect differences between domains and sectors.

- By performing and discussing internal consistency tests on respondents' scores, the authors now analyze item covariance for purposes of identifying extraneous and redundant items (though it appears that none were found).

We thank the reviewer for the comment. We implemented a factor analysis and 38 items have been excluded from the analysis. More explanation has been provided in our response to the reviewer comment 1 above.

- A further copy-edit of the manuscript, particularly of newly added text, is necessary.

We thank the reviewer for the comment. A copy-edit of the manuscript has been completed.

VERSION 3 – REVIEW

REVIEWER	Bloom, David Harvard School of Public Health, Department of Global Health and Population
REVIEW RETURNED	21-Jan-2023

GENERAL COMMENTS	The reviewer applauds the substantial improvements the authors have made to the manuscript. The authors thoroughly revised their tests of questionnaire items' internal validity and reliability. They streamlined their battery of tests from six to two and raised the threshold test scores required for item retention. These changes led the authors to remove 38 items from their analysis. The reviewer commends the authors for their diligence in undertaking these revisions, which bolster the credibility of their results. The authors also provided welcome clarity about the computation of domain, sector, and country scores. While domain and sector scores are unweighted averages of respondents' ratings of retained items, country scores are weighted averages of sector scores, with sector weights proportional to the share of respondents in a sector. The reviewer would have opted for unweighted country scores—the authors effectively assign each country a different set of sector weights based rather arbitrarily on the distribution of respondents across sectors in each country—but defers to the authors' discretion and appreciates their clarifications in any case. The authors significantly reorganized the description of their results. The reviewer emphatically approves of the authors' decision to present results by country rather than by sector and domain. The reorganized results section makes punchy and meaningful comparisons with clear relevance to national policymakers. The authors also make available an appendix delineating the origins of questionnaire items that came from preexisting effort indices and indicating sources of inspiration for other items. This appendix is a valuable addition that interested researchers may find quite useful. The reviewer therefore thanks the authors for their efforts. The reviewer wishes to revisit only one major comment raised in previous reviews—that of heterogeneity—to resolve a misunderstanding. The authors introduced heterogeneity tests in response to the reviewer's concern that respondents' domains of expertise may shape their outlooks in systematic (i.e., cross-country) ways, a possibility that matters because of high cross-country variability in respondents' domains of expertise. The presentation of these heterogeneity tests in the initial revised
---

	manuscript and its annexed materials gave the reviewer a seemingly incorrect impression of the concept being tested. The reviewer believed the authors had grouped respondents by country and own domain of expertise, regardless of sector, and tested for heterogeneity the scores that respondents in each group had given to their respective sectors, despite these sectors being different. In their responses to reviewer comments attached to the second revised manuscript, the authors clarified that their allusions to “domains” refer not to respondents’ own domains of expertise, but to the ratings they assigned to each domain (including and in addition to their own). The reviewer now understands that the authors grouped respondents by country and sector, with all domains of expertise notionally represented in each group, and tested for heterogeneity the ratings that respondents in each group gave to items in each domain. If the reviewer’s current understanding is correct, then the authors’ heterogeneity tests do indeed respond to the reviewer’s concern, and the generally low scores that resulted from these tests assuage the reviewer. The reviewer recommends that the authors incorporate a thorough clarification into the manuscript itself. The authors’ latest revisions to the manuscript still leave ample room for misinterpretation by readers. It would be sufficient if the authors replaced first two sentences of the paragraph about heterogeneity testing on page 7 (“Given that the measures... using meta-analytical techniques”) with a more detailed description of the possible problem—that respondents with differing domains of expertise assigned systematically different ratings to the same items, and that respondents’ domains of expertise are distributed differently across countries—and the precise groupings (by country and sector) within which respondents’ ratings were compared, as well as the ratings (domain-level simple averages?) on which intra-group heterogeneity was assessed. The reviewer believes these sorts of modifications would not only dispel confusion, but also help convince readers of the credibility of the authors’ findings. The following are a few minor comments:  • The first paragraph of page 9 reads, “In all countries, except Tanzania, the majority of respondents were from the public sector and the services and programs domains.” Examination of Table 1 suggests that Ethiopia, not Tanzania, is the only country with minorities of respondents from the public sector and from the domain of services and programs. Tanzania appears to have a majority of respondents from the services and programs domain. • On page 11, the discussion of the lowest scores in Senegal reads, “The lowest scores recorded were in civil society roles in LM (4.3, 95% CI:2.7-5.8), advocacy (4.5, 95% CI:2.8-6.2), research and in services and programs (4.8, 95% CI:3.4-6.3).” The reviewer recommends placing “LM” either before or after listing the four lowest-scoring domains (which happen to be in the labor market sector). • The second paragraph of page 12 features an abrupt swing from negativity about women’s prospects of attaining senior positions (“These findings agreed with previous predictions by the World Economic Forum that women may have to wait for 100 years to pair males in leadership roles in bigger institutions and take up ministerial roles”) to positivity about Rwanda’s example as a leader on gender equality (“This is currently seen in the Global Gender Gap (GGG) Report, which ranks Rwanda in the top ten countries
--	--

	with the smallest gender gap among countries worldwide”). Consider inserting language to ease this transition.
--	--

VERSION 3 – AUTHOR RESPONSE

Reviewer: 1

Dr. David Bloom, Harvard School of Public Health

Comments to the Author:

The reviewer applauds the substantial improvements the authors have made to the manuscript.

The authors thoroughly revised their tests of questionnaire items' internal validity and reliability. They streamlined their battery of tests from six to two and raised the threshold test scores required for item retention. These changes led the authors to remove 38 items from their analysis. The reviewer commends the authors for their diligence in undertaking these revisions, which bolster the credibility of their results.

The authors also provided welcome clarity about the computation of domain, sector, and country scores. While domain and sector scores are unweighted averages of respondents' ratings of retained items, country scores are weighted averages of sector scores, with sector weights proportional to the share of respondents in a sector. The reviewer would have opted for unweighted country scores—the authors effectively assign each country a different set of sector weights based rather arbitrarily on the distribution of respondents across sectors in each country—but defers to the authors' discretion and appreciates their clarifications in any case.

The authors significantly reorganized the description of their results. The reviewer emphatically approves of the authors' decision to present results by country rather than by sector and domain. The reorganized results section makes punchy and meaningful comparisons with clear relevance to national policymakers.

The authors also make available an appendix delineating the origins of questionnaire items that came from preexisting effort indices and indicating sources of inspiration for other items. This appendix is a valuable addition that interested researchers may find quite useful. The reviewer therefore thanks the authors for their efforts.

The reviewer wishes to revisit only one major comment raised in previous reviews—that of heterogeneity—to resolve a misunderstanding. The authors introduced heterogeneity tests in response to the reviewer's concern that respondents' domains of expertise may shape their outlooks in systematic (i.e., cross-country) ways, a possibility that matters because of high cross-country variability in respondents' domains of expertise. The presentation of these heterogeneity tests in the initial revised manuscript and its annexed materials gave the reviewer a seemingly incorrect impression of the concept being tested. The reviewer believed the authors had grouped respondents by country and own domain of expertise, regardless of sector, and tested for heterogeneity the scores that respondents in each group had given to their respective sectors, despite these sectors being different. In their responses to reviewer comments attached to the second revised manuscript, the authors clarified that their allusions to “domains” refer not to respondents' own domains of expertise, but to the ratings they assigned to each domain (including and in addition to their own). The reviewer now understands that the authors grouped respondents by country and sector, with all domains of expertise notionally represented in each group, and tested for heterogeneity the ratings that respondents in each group gave to items in each domain. If the reviewer's current understanding is

correct, then the authors' heterogeneity tests do indeed respond to the reviewer's concern, and the generally low scores that resulted from these tests assuage the reviewer.

We thank the reviewer for their careful consideration and appreciate the request for clarification. The reviewer's understanding described above is correct.

The reviewer recommends that the authors incorporate a thorough clarification into the manuscript itself. The authors' latest revisions to the manuscript still leave ample room for misinterpretation by readers. It would be sufficient if the authors replaced first two sentences of the paragraph about heterogeneity testing on page 7 ("Given that the measures... using meta-analytical techniques") with a more detailed description of the possible problem—that respondents with differing domains of expertise assigned systematically different ratings to the same items, and that respondents' domains of expertise are distributed differently across countries—and the precise groupings (by country and sector) within which respondents' ratings were compared, as well as the ratings (domain-level simple averages?) on which intra-group heterogeneity was assessed. The reviewer believes these sorts of modifications would not only dispel confusion, but also help convince readers of the credibility of the authors' findings.

We thank the reviewer for this comment. We have made a revision to reflect this comment and have made changes to the description of heterogeneity testing in the analysis section (p.7) which now reads:

Heterogeneity testing

Because respondents with different levels of expertise may have systematically assigned different ratings to the same items and the domains of expertise of respondents were differently distributed across countries, we checked for heterogeneity across domains using meta-analytical techniques. This analysis used pooled data by country to allow for sufficient sample size. To assess the overall extent of heterogeneity and whether it was adequate to compute summary estimates, the fixed-effects were compared to random-effect estimates. By using this method, substantial discrepancies between the estimates from the two models indicate considerable heterogeneity that can make summary estimates misleading.[36] To further assess heterogeneity across domains in respective countries I-square (I²) and Tau-squared were used. The I² values of 25%, 50%, and 75% have been interpreted as representing small, moderate, and high levels of heterogeneity.[37] Tau-squared captures the variance between measurements of efforts by domains which can be captured by random-effect meta-analytic models.

The following are a few minor comments:

- The first paragraph of page 9 reads, "In all countries, except Tanzania, the majority of respondents were from the public sector and the services and programs domains." Examination of Table 1 suggests that Ethiopia, not Tanzania, is the only country with minorities of respondents from the public sector and from the domain of services and programs. Tanzania appears to have a majority of respondents from the services and programs domain.

We thank the reviewer for this comment and agree with the reviewer's observation. We have made a revision to reflect this comment and the paragraph now reads:

In all countries, except Ethiopia, the majority of respondents were from the public sector and the services and programs domains.

- On page 11, the discussion of the lowest scores in Senegal reads, “The lowest scores recorded were in civil society roles in LM (4.3, 95% CI:2.7-5.8), advocacy (4.5, 95% CI:2.8-6.2), research and in services and programs (4.8, 95% CI:3.4-6.3).” The reviewer recommends placing “LM” either before or after listing the four lowest-scoring domains (which happen to be in the labor market sector).

We thank the reviewer for this recommendation and agree with the reviewer’s observation. We have made a revision to reflect this comment and this section of the discussion now reads:

The lowest scores were recorded across domains in LM, civil society (4.3, 95% CI:2.7-5.8), advocacy (4.5, 95% CI:2.8-6.2), research (4.7, 95% CI:3.2-6.3) and in services and programs (4.8, 95% CI:3.4-6.3). The scores for WE were consistently around 6 or higher except in research (5.7, 95% CI:4.5-6.9).

- The second paragraph of page 12 features an abrupt swing from negativity about women’s prospects of attaining senior positions (“These findings agreed with previous predictions by the World Economic Forum that women may have to wait for 100 years to pair males in leadership roles in bigger institutions and take up ministerial roles”) to positivity about Rwanda’s example as a leader on gender equality (“This is currently seen in the Global Gender Gap (GGG) Report, which ranks Rwanda in the top ten countries with the smallest gender gap among countries worldwide”). Consider inserting language to ease this transition.

We thank the reviewer for this recommendation and agree with the reviewer’s observation. We have made a revision to reflect this comment and the second paragraph of the discussion section (p.10-11) now reads:

The DDEI scores revealed patterns that were helpful to guide discussions regarding needed improvements in respective countries across five domains: namely, policymaking, services and programs, advocacy, research, and civil society. The patterns of efforts scores for the LM, WE, FP, GEI and ED sectors suggested great challenges for women’s self-determination and empowerment, employment, and leadership. These findings agreed with previous predictions by the World Economic Forum that women may have to wait for 100 years to pair males in leadership roles in bigger institutions and take up ministerial roles. However, in some countries remarkable progress has been recorded indicating potential opportunity to attain gender parity much sooner. This is currently seen in the Global Gender Gap (GGG) Report, which ranks Rwanda in the top ten countries with the smallest gender gap among countries worldwide.[38, 39]

Reviewer: 1

Competing interests of Reviewer: No competing interests.